# ON IDENTIFIABILITY IN TRANSFORMERS

**Gino Brunner**[1*]**, Yang Liu**[2*]**, Damián Pascual**[1*]**, Oliver Richter**[1]**,**
**Massimiliano Ciaramita**[3]**, Roger Wattenhofer**[1]
Departments of [1]Electrical Engineering and Information Technology, [2]Computer Science
ETH Zurich, Switzerland
[3]Google Research, Zurich, Switzerland
[1]{brunnegi,dpascual,richtero,wattenhofer}@ethz.ch,
[2]liu.yang@alumni.ethz.ch
[3]massi@google.com

## ABSTRACT

In this paper we delve deep in the Transformer architecture by investigating two of its core components: self-attention and contextual embeddings. In particular, we study the identifiability of attention weights and token embeddings, and the aggregation of context into hidden tokens. We show that, for sequences longer than the attention head dimension, attention weights are not identifiable. We propose *effective attention* as a complementary tool for improving explanatory interpretations based on attention. Furthermore, we show that input tokens retain to a large degree their identity across the model. We also find evidence suggesting that identity information is mainly encoded in the angle of the embeddings and gradually decreases with depth. Finally, we demonstrate strong mixing of input information in the generation of contextual embeddings by means of a novel quantification method based on gradient attribution. Overall, we show that self-attention distributions are not directly interpretable and present tools to better understand and further investigate Transformer models.

## 1 INTRODUCTION

In this paper we investigate neural models of language based on self-attention by concentrating on the concept of *identifiability*. Intuitively, identifiability refers to the ability of a model to learn stable representations. This is arguably a desirable property, as it affects the replicability and interpretability of the model's predictions. Concretely, we focus on two aspects of identifiability. The first is related to *structural identifiability* (Bellman & Åström, 1970): the theoretical possibility (a priori) to learn a unique optimal parameterization of a statistical model. From this perspective, we analyze the identifiability of attention weights, what we call *attention identifiability*, in the self-attention components of transformers (Vaswani et al., 2017), one of the most popular neural architectures for language encoding and decoding. We also investigate *token identifiability*, as the fine-grained, word-level mappings between input and output generated by the model. The role of attention as a means of recovering input-output mappings, and various types of explanatory insights, is currently the focus of much research and depends to a significant extent on both types of identifiability.

We contribute the following findings to the ongoing work: With respect to attention indentifiability, in Section 3, we show that – under mild conditions with respect to input sequence length and attention head dimension – the attention weights for a given input are not identifiable. This implies that there can be infinitely many different attention weights that yield the same output. This finding challenges the direct interpretability of attention distributions. As a supplement, we propose the concept of *effective attention*, a diagnostic tool that examines attention weights for model explanations by removing the weight components that do not influence the model's predictions.

---

[*]Equal contribution with authors in alphabetical order. Yang Liu initiated the transformer models study, perceived and performed the study of attention identifiability and effective attention, i.e., Section 3 and Appendix A, and contributed to the token attribution discussions and calculations.

With respect to token identifiability, in Section 4, we devise an experimental setting where we probe the hypothesis that contextual word embeddings maintain their identity as they pass through successive layers of a transformer. This is an assumption made in much current research, which has not received a clear validation yet. Our findings give substance to this assumption, although it does not always hold in later layers. Furthermore, we show that the identity information is largely encoded in the angle of the embeddings and that it can be recovered by a nearest neighbour lookup after a learned linear mapping from hidden to input token space.

In Section 5 we further investigate the contribution of all input tokens in the generation of the contextual embeddings in order to quantify the mixing of token and context information. We introduce *Hidden Token Attribution*, a quantification method based on gradient attribution. We find that self-attention strongly mixes context and token contributions. Token contribution decreases monotonically with depth, but the corresponding token typically remains the largest individual contributor. We also find that, despite visible effects of long term dependencies, the context aggregated into the hidden embeddings is mostly local. We notice how, remarkably, this must be an effect of learning.

## 2 BACKGROUND ON TRANSFORMERS

The Transformer (Vaswani et al., 2017) is currently the neural architecture of choice for natural language processing (NLP). At its core it consists of several multi-head self-attention layers. In these layers, every token of the input sequence attends to all other tokens by projecting its embedding to a query, key and value vector. Formally, let $Q \in \mathbb{R}^{d_s \times d_q}$ be the query matrix, $K \in \mathbb{R}^{d_s \times d_q}$ the key matrix and $V \in \mathbb{R}^{d_s \times d_v}$ the value matrix, where $d_s$ is the sequence length and $d_q$ and $d_v$ the dimension of query and value vectors, respectively. The output of an attention head is given by:

$$\text{Attention}(Q, K, V) = A \cdot V \qquad \text{with} \quad A = \text{softmax}\left(\frac{QK^T}{\sqrt{d_q}}\right) \qquad (1)$$

The attention matrix $A \in \mathbb{R}^{d_s \times d_s}$ calculates, for each token in the sequence, how much the computation of the hidden embedding at this sequence position should be influenced by each of the other (hidden) embeddings. Self-attention is a non-local operator, which means that at any layer a token can attend to all other tokens regardless of the distance in the input. Self-attention thus produces so-called *contextual word embeddings*, as successive layers gradually aggregate contextual information into the embedding of the input word.

We focus on a Transformer model called BERT (Devlin et al., 2019), although our analysis can be easily extended to other models such as GPT, (Radford et al., 2018; 2019) RoBERTa (Liu et al., 2019), XLNet (Yang et al., 2019b), or ALBERT (Lan et al., 2020). BERT operates on input sequences of length $d_s$. We denote input tokens in the sentence as $x_i$, where $i \in [1, ..., d_s]$. We use $x_i \in \mathbb{R}^d$ with embedding dimension $d$ to refer to the sum of the token-, segment- and position embeddings corresponding to the input word at position $i$. We denote the contextual embedding at position $i$ and layer $l$ as $e_i^l$. Lastly, we refer to the inputs and embeddings of all sequence positions as matrices $X$ and $E$, respectively, both in $\mathbb{R}^{d_s \times d}$. For all experiments we use the pre-trained uncased BERT-Base model as provided by Devlin et al. (2019)[1].

## 3 ATTENTION IDENTIFIABILITY

We begin with the identifiability analysis of self-attention weights. Drawing an analogy with structural identifiability (Bellman & Åström, 1970), we state that the attention weights of an attention head for a given input are *identifiable* if they can be uniquely determined from the head's output.[2] We emphasize that attention weights are input dependent and *not model parameters*. However, their identifiability affects the interpretability of the output, i.e., whether attention weights can provide the basis for explanatory insights on the model's predictions (cf. Jain & Wallace (2019) and Wiegreffe & Pinter (2019)). If attention is not identifiable, explanations based on attention may be unwarranted.

---

[1] https://github.com/google-research/bert
[2] Cf. Appendix A.1 for more background on indentifiability.

The output of a multi-head attention layer is the summation over each of the $h$ single head outputs (cf. Eq. 1) multiplied by the matrix $\boldsymbol{H} \in \mathbb{R}^{d_v \times d}$ with reduced head dimension $d_v = d/h$,

$$\text{Attention}(\boldsymbol{Q}, \boldsymbol{K}, \boldsymbol{V})\boldsymbol{H} = \boldsymbol{AEW}^V \boldsymbol{H} = \boldsymbol{AT} \tag{2}$$

where $\boldsymbol{W}^V \in \mathbb{R}^{d \times d_v}$ projects the embedding $\boldsymbol{E}$ into the value matrix $\boldsymbol{V} = \boldsymbol{EW}^V$, and we define $\boldsymbol{T} = \boldsymbol{EW}^V \boldsymbol{H}$. Here, the layer and head indices are omitted for simplicity, since the proof below is valid for each individual head and layer in Transformer models. Intuitively, the head output is a linear combination of the $\boldsymbol{T}$ vectors using the attention as weighting coefficients. If the sequence length, i.e. the number of weighting coefficient, is larger than the rank of $\boldsymbol{T}$, attention weights are not uniquely determined from the head output; i.e., they include free variables. In other words, some of the $\boldsymbol{T}$ rows are linear combinations of others. We now prove, by analyzing the null space dimension of $\boldsymbol{T}$, that attention weights are not identifiable using the head or final model output.

## 3.1 Upper Bound for rank$(T)$

We first derive the upper bound of the rank of matrix $\boldsymbol{T} = \boldsymbol{EW}^V \boldsymbol{H}$. Note that $\text{rank}(\boldsymbol{ABC}) \leq \min(\text{rank}(\boldsymbol{A}), \text{rank}(\boldsymbol{B}), \text{rank}(\boldsymbol{C}))$, therefore,

$$\begin{aligned} \text{rank}\,(\boldsymbol{T}) &\leq \min\left(\text{rank}(\boldsymbol{E}), \text{rank}(\boldsymbol{W}^V), \text{rank}(\boldsymbol{H})\right) \\ &\leq \min(d_s, d, d, d_v, d_v, d) \\ &= \min\,(d_s, d_v)\,. \end{aligned} \tag{3}$$

The second step holds since $\text{rank}(\boldsymbol{E}) \leq \min(d_s, d)$, $\text{rank}(\boldsymbol{W}^V) \leq \min(d, d_v)$ and $\text{rank}(\boldsymbol{H}) \leq \min(d_v, d)$.

## 3.2 The null space of $T$

The (left) null space $\text{LN}(\boldsymbol{T})$ of $\boldsymbol{T}$ describes all vectors that are mapped to the zero vector by $\boldsymbol{T}$:

$$\text{LN}(\boldsymbol{T}) = \{\tilde{\boldsymbol{x}}^T \in \mathbb{R}^{1 \times d_s} | \tilde{\boldsymbol{x}}^T \boldsymbol{T} = \boldsymbol{0}\} \tag{4}$$

Its special property is that, for $\tilde{\boldsymbol{A}} = [\tilde{\boldsymbol{x}}_1, \tilde{\boldsymbol{x}}_2, ..., \tilde{\boldsymbol{x}}_{d_s}]^T$ where $\tilde{\boldsymbol{x}}_i^T$ are vectors in this null space,

$$(\boldsymbol{A} + \tilde{\boldsymbol{A}})\boldsymbol{T} = \boldsymbol{AT}. \tag{5}$$

If the dimension of $\text{LN}(\boldsymbol{T})$ is not zero, there exist infinitely many attention weights $\boldsymbol{A} + \tilde{\boldsymbol{A}}$ yielding the exact same attention layer output and final model outputs. By applying the Rank-Nullity theorem, the dimension of the null space is:

$$\dim(\text{LN}(\boldsymbol{T})) = d_s - \text{rank}\,(\boldsymbol{T}) \geq d_s - \min\,(d_s, d_v) = \begin{cases} d_s - d_v, & \text{if } d_s > d_v \\ 0, & \text{otherwise} \end{cases} \tag{6}$$

here we make use of the facts that $\dim(\text{LN}(\boldsymbol{T})) = \dim(\text{N}(\boldsymbol{T}^T))$ and $\text{rank}(\boldsymbol{T}) = \text{rank}(\boldsymbol{T}^T)$ where $\text{N}(\boldsymbol{T})$ represents the null space of a matrix $\boldsymbol{T}$. Equality holds if $\boldsymbol{E}$, $\boldsymbol{W}^V$ and $\boldsymbol{H}$ are of full rank and their matrix product does not bring further rank reductions.

Hence, when the sequence length is larger than the attention head dimension ($d_s > d_v$), self-attention is not unique. Furthermore, the null space dimension increases with the sequence length. In the next section we show that the attention weights are also non-identifiable; i.e., the non-trivial null space of $\boldsymbol{T}$ exists, even when the weights are constrained within the probability simplex.

## 3.3 The null space with probability constraints

Since $\boldsymbol{A}$ is the result of a softmax operation, its rows are constrained within the probability simplex: $\boldsymbol{A} \geq 0$ (element-wise), and $\boldsymbol{A1} = \boldsymbol{1}$, where $\boldsymbol{1} \in \mathbb{R}^{d_s}$ is the vector of all ones. However, the derivation in Section (3.2) does not take these constraints into account. It shows that $\boldsymbol{A}$ is not unique, but it does not prove that alternative attention weights exist within the probability simplex, and thus that $\boldsymbol{A}$ is not identifiable. Below, we show that $\tilde{\boldsymbol{A}}$ exists in $\text{LN}(\boldsymbol{T})$ also when constraining the weights of $(\boldsymbol{A} + \tilde{\boldsymbol{A}})$ to the probability simplex.

For the row vectors from an alternative attention matrix $\boldsymbol{A} + \tilde{\boldsymbol{A}}$ to be valid distributions, we require, in addition to $\boldsymbol{A} \geq \boldsymbol{0}$ (element-wise) and $\boldsymbol{A1} = \boldsymbol{1}$, that $\boldsymbol{A} + \tilde{\boldsymbol{A}} \geq \boldsymbol{0}$ (element-wise), and $\boldsymbol{A1} + \tilde{\boldsymbol{A}}\boldsymbol{1} = \boldsymbol{1}$. Furthermore, $\tilde{\boldsymbol{A}}$ must be in the (left) null space of $\boldsymbol{T}$. We formalize the three conditions below:

$$\text{a) } \tilde{\boldsymbol{A}}\boldsymbol{T} = \boldsymbol{0} \quad \text{b) } \tilde{\boldsymbol{A}}\boldsymbol{1} = \boldsymbol{0} \quad \text{c) } \tilde{\boldsymbol{A}} \geq -\boldsymbol{A} \tag{7}$$

Conditions (7a) and (7b) can be combined as $\boldsymbol{A}[\boldsymbol{T}, \boldsymbol{1}] = \boldsymbol{0}$, where $[\boldsymbol{T}, \boldsymbol{1}]$ is the augmented matrix resulting from adding a column vector of ones to $\boldsymbol{T}$. Reusing the argument presented in Section (3.2), the dimension of its (left) null space is: $\dim(\text{LN}([\boldsymbol{T}, \boldsymbol{1}])) \geq \max(d_s - d_v - 1, 0)$. Thus, for $d_s - d_v > 1$, the null space of $[\boldsymbol{T}, \boldsymbol{1}]$ exists: it is a linear subspace of $\text{LN}(\boldsymbol{T})$.

We now prove that condition (7c) can also be satisfied. We begin by providing an intuitive justification. The condition restricts the space of $\tilde{\boldsymbol{A}}$ from $\text{LN}([\boldsymbol{T}, \boldsymbol{1}])$ to be a bounded region which could be different for each row vector $\boldsymbol{a} = (a_1, a_2, \ldots)$ of $\boldsymbol{A}$. The null space $\text{LN}([\boldsymbol{T}, \boldsymbol{1}])$ contains $\boldsymbol{0}$, defining a surface passing through the origin. Since $\boldsymbol{a}$ is a probability vector, resulting from a softmax transformation, each of its components is strictly positive, i.e., $\boldsymbol{A} > \boldsymbol{0}$ (element-wise). Hence, there exists $\epsilon > 0$ such that any point $\tilde{\boldsymbol{a}}$ in the sphere centered at the origin with radius $\epsilon$ will satisfy condition (7c), $\tilde{\boldsymbol{a}} > -\boldsymbol{a}$. Crucially, this sphere intersects the null space, as they share the origin. Any point in this intersection satisfies all three conditions in (7).

Formally, the construction of the null space vector $\tilde{\boldsymbol{a}}$ for the alternative attention weights $\tilde{\boldsymbol{a}} + \boldsymbol{a}$ goes as follows. For a vector $\tilde{\boldsymbol{a}} = (\tilde{a}_1, \tilde{a}_2, ...) \in \text{LN}([\boldsymbol{T}, \boldsymbol{1}])$, to ensure one of its negative components $\tilde{a}_i < 0$ satisfying condition (7c) that $\tilde{a}_i \geq -a_i$, one could shrink its magnitude into $\lambda \tilde{\boldsymbol{a}}$ with $0 \leq \lambda \leq -a_i/\tilde{a}_i$, so $\lambda \tilde{a}_i \geq -a_i$. Considering all negative components $i$, the overall scaling factor is $\lambda_{max} = \min_{i \in \{i|\tilde{a}_i < 0\}}(-a_i/\tilde{a}_i)$ so that the direction from the origin $\{\lambda \tilde{\boldsymbol{a}}|0 \leq \lambda \leq \lambda_{max}\}$ satisfies condition (7c). Here $\lambda_{max}$ is strictly greater than 0 because $a_i > 0$. Only when there exists an index $i$ that $\tilde{a}_i < 0$ and $a_i \approx 0$, then this particular null space direction $\tilde{\boldsymbol{a}}$ is highly confined. In the extreme case, where $\boldsymbol{a}$ is a one-hot distribution, the solution should be an $\tilde{\boldsymbol{a}}$ with only one negative component. If such an $\tilde{\boldsymbol{a}}$ does not exist in $\text{LN}([\boldsymbol{T}, \boldsymbol{1}])$, the solution collapses to the trivial single point $\tilde{\boldsymbol{a}} = \boldsymbol{0}$. However, in general, $\text{LN}([\boldsymbol{T}, \boldsymbol{1}])$ with probability constraints is non-trivial.

## 3.4 EFFECTIVE ATTENTION

The non-identifiability of self-attention, due to the existence of the non-trivial null space of $\boldsymbol{T}$, challenges the interpretability of attention weights. However, one can decompose attention weights $\boldsymbol{A}$ into the component in the null space $\boldsymbol{A}^{\|}$ and the component orthogonal to the null space $\boldsymbol{A}^{\perp}$:

$$\boldsymbol{A}\boldsymbol{T} = (\boldsymbol{A}^{\|} + \boldsymbol{A}^{\perp})\boldsymbol{T} = \boldsymbol{A}^{\perp}\boldsymbol{T} \tag{8}$$

since $\boldsymbol{A}^{\|} \in \text{LN}(\boldsymbol{T}) \implies \boldsymbol{A}^{\|}\boldsymbol{T} = \boldsymbol{0}$. Hence, we propose a novel concept named *effective attention*,

$$\boldsymbol{A}^{\perp} = \boldsymbol{A} - \text{Projection}_{\text{LN}(T)}\boldsymbol{A}, \tag{9}$$

which is the part of the attention weights that actually affects the model output. The null space projection is calculated by projecting attention weights into the left null space basis, i.e., the associated left singular vectors.

Here the definition of effective attention uses $\text{LN}(\boldsymbol{T})$ instead of the null space $\text{LN}([\boldsymbol{T}, 1])$ with probability constraints. As a consequence, effective attention is not guaranteed to be a probability distribution; e.g., some of the weights might be negative. One could define effective attention as the minimal norm alternative attention using $\text{LN}([\boldsymbol{T}, \boldsymbol{1}])$, or possibly constrain it within the probability simplex. However, in this case, the minimal norm alternative attention is not orthogonal to the null space $\text{LN}(\boldsymbol{T})$ anymore. It seems unclear how to interpret the minimal norm alternative attention. The reason being that the distinction between components that affect or do not affect the output computation[3] are defined with respect to $\text{LN}(\boldsymbol{T})$ and not with respect to $\text{LN}([\boldsymbol{T}, \boldsymbol{1}])$. In fact, there may be useful information in the sign of the effective attention components. Although the combination of value vectors in the transformer architecture uses weights in the probability simplex, these probability constraints on attention may not be necessary. Hence, we provide here the base version of an effective attention, and leave the investigation of other formulations for future research.

---

[3] A key aspect of the concept of identifiability.

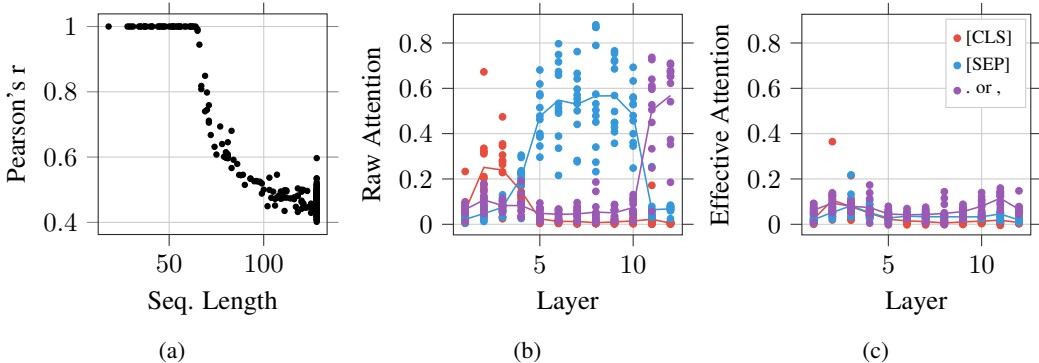

Figure 1: (a) Each point represents the Pearson correlation coefficient of effective attention and raw attention as a function of token length. (b) Raw attention vs. (c) effective attention, where each point represents the average (effective) attention of a given head to a token type.

### 3.5 EMPIRICAL EVIDENCE

We conclude by providing some initial empirical evidence in support of the notion that effective attention can serve as a complementary diagnostic tool for examining how attention weights influence model outputs.

First, we show that effective attention can be detected, and can diverge significantly from raw attention. In Figure 1a, we illustrate how the Pearson correlation between effective and raw attention decreases with sequence length. We use the same Wikipedia samples as in Clark et al. (2019) with maximum sequence length 128. This result is in line with our theoretical finding in Eq. 6 that states an increase in the dimension of the null space with the sequence length. Given a bigger null space, more of the raw attention becomes irrelevant, yielding a lower correlation between effective and raw attention. Notice how, for sequences with fewer than $d_v = 64$ tokens, the associated null space dimension is zero, and hence attention and effective attention are identical (Pearson correlation of value 1). This loss of correlation with increased sequence length questions the use of attention as explanation in practical models, where it is not uncommon to use large sequence lengths. A few examples include: BERT for question answering (Alberti et al., 2019) and XL-Net (Yang et al., 2019a) with $d_s = 512$, or document translation (Junczys-Dowmunt, 2019) with $d_s = 1000$.

To illustrate the point further, Figure 1 shows in (b) raw attention $\boldsymbol{A}$ and (c) effective attention $\boldsymbol{A}^\perp$, using again the data of Clark et al. (2019). We compute the average attention of BERT and compare it to the corresponding average effective attention. Clark et al. (2019) conclude that the [CLS] token attracts more attention in early layers, the [SEP] tokens attract more in middle layers, and periods and commas do so in deep layers. However, after a gradient based investigation, they propose that attention to the [SEP] token is generally a "no-op". The effective attention weights suggest a more consistent pattern: while periods and commas seem to generally attract more attention than [CLS] and [SEP], the peak of [SEP] token observed by raw attention has disappeared. Effective attention provides an explanation: the [SEP] token peak is irrelevant to the computation of the output for middle layers; i.e., it is in the null space component of the corresponding attention vector. The same arguments also hold for the sharp peak of raw attention on punctuation tokens between layers 10 and 12. An additional example showing similar results can be found in Appendix A.2. See also Appendix A.3, where we discuss in more depth a case where effective attention would support interpretive conclusions that differ from those one might draw solely based on raw attention. In conclusion, effective attention can help discover interesting interactions encoded in the attention weights which may be otherwise obfuscated by the null attention.

## 4 TOKEN IDENTIFIABILITY

We now study the other fundamental element of transformers; the hidden vector representations of tokens, or contextual word embeddings. It is commonly assumed that a contextual word embedding

keeps its "identity", which is tied to the input word, as it passes through the self-attention layers. Specifically, we identify three cases where this assumption is made implicitly without justification.

- Visualizations/interpretations linking attention weights to attention between words, when in fact the attention is between embeddings, i.e., mixtures of multiple words (Vaswani et al., 2017; Devlin et al., 2019; Vig, 2019; Clark et al., 2019; Raganato & Tiedemann, 2018; Voita et al., 2019; Tang et al., 2018; Wangperawong, 2018; Padigela et al., 2019; Baan et al., 2019; Dehghani et al., 2019; Zenkel et al., 2019).
- Attention accumulation methods that sum the attention to a specific sequence position over layers and/or attention heads, when the given position might encode a different mixture of inputs in each layer (Clark et al., 2019; Baan et al., 2019; Klein & Nabi, 2019; Coenen et al., 2019).
- Using classifiers to probe hidden embeddings for word-specific aspects without factoring in how much the word is still represented (Lin et al., 2019; Peters et al., 2018).

To investigate this assumption we introduce the concept of *token identifiability*, as the existence of a mapping assigning contextual embeddings to their corresponding input tokens. Formally, we state that an embedding $e_i^l$ is identifiable if there exists a classification function $c(\cdot)$ such that $c(e_i^l) = x_i$. For identifiability we only require $c(e_i^l)$ to recover $x_i$ in a nearest neighbour sense within the same input sentence. Therefore, for each layer $l$ we define $c_l(\cdot) = NN(g_l(\cdot))$, where $NN$ is a 1-nearest neighbour lookup, and $g_l : \mathbb{R}^d \to \mathbb{R}^d$ is a continuous function mapping embeddings to vectors of real numbers. Since we cannot prove the existence of $g_l$ analytically, we instead use a function approximator $\hat{g}_l(e_i^l) = \hat{x}_i$, trained on a dataset of $(e_i^l, x_i)$ pairs. We then say that $e_i^l$ is identifiable if $\hat{c}_l(e_i^l) = NN(\hat{g}_l(e_i^l)) = x_i$. For evaluation we report the *token identifiability rate* defined as the percentage of correctly identified tokens.

## 4.1 SETUP

For the experiments in this and subsequent sections we use the development dataset from the Microsoft Research Paraphrase Corpus (MRPC) dataset (Dolan & Brockett, 2005), while in Appendix D we provide results on two additional datasets. The MRPC development set contains 408 examples with a sequence length $d_s$ between 26 and 92 tokens, with 58 tokens on average. We pass all 408 sentences (21,723 tokens) through BERT and extract for each token the input embeddings $x_i$ and the hidden embeddings $e_i^l$ at all layers. We then train $\hat{g}$ on the regression task of predicting input tokens $x_i$ from hidden tokens $e_i^l$. We experiment with two loss functions and similarity measures for finding the nearest neighbour; cosine distance and $L_2$ distance. We use 10-fold cross validation with 70/15/15 train/validation/test splits per fold and ensure that tokens from the same sentence are not split across sets. The validation set is used for early stopping. See Appendix B.1 for details.

## 4.2 EXPERIMENTAL RESULTS AND DISCUSSION

In a first experiment, we use a linear perceptron without bias and a non-linear MLP $\hat{g}_l^{MLP}$, where training, validation and test data all come from layer $l$. Figure 2a shows the test set token identifiability rate of $\hat{c}_l$ for $l = [1, ..., 12]$. We also report a naive baseline $\hat{g}_l^{naive}(e_i^l) = e_i^l$, i.e., we directly retrieve the nearest neighbour of $e_i^l$ from the input tokens. The results for $\hat{g}_l^{naive}$ show that, according to both similarity measures, contextual embeddings in BERT stay close to their input embeddings up to layer 4, followed by a linear decrease in token identifiability rate. By training a transformation to recover the original embedding, we see that most of the identity information is still present in the contextualized embeddings. Specifically, a linear projection is enough to recover 93% of the tokens in the last layer based on a cosine distance nearest neighbour lookup.

This experiment shows that although the identifiablity rate decreases with depth, tokens remain mostly identifiable across layers. Furthermore, we find that cosine distance is more effective to recover identity than $L_2$ distance. Therefore, we conjecture that BERT encodes most of the identity information in the angle of the embeddings. Finally, Lin et al. (2019) show that BERT discards much of the positional information after layer 3. However, tokens remain largely identifiable throughout the model, indicating that BERT does not only rely on the positional embeddings to track token identity. To provide further insights into contextual word embeddings, Appendix B.5 shows results for recovering neighbouring input tokens.

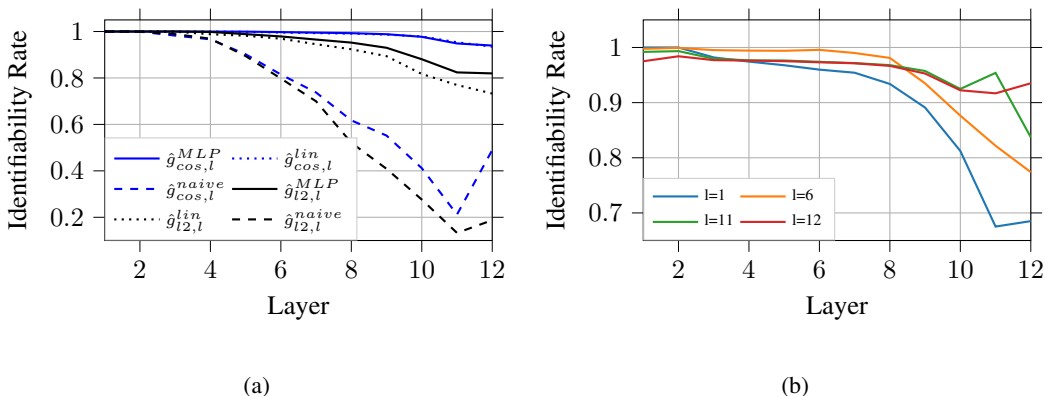

(a)     (b)

Figure 2: (a) Identifiability of contextual word embeddings at different layers. Here, $\hat{g}$ is trained and tested on the same layer. (b) $g_{cos,l}^{lin}$ trained on layer $l$ and tested on all layers.

In a second experiment we test how well the $\hat{g}_{cos,l}^{lin}$ trained only on $(e_i^l, x_i)$ pairs from one layer $l$ generalizes to all layers, see Figure 2b. For $l = 1$, the token identifiability rate on subsequent layers drops quickly to below 70% at layers 11 and 12. Interestingly, for $l = 12$ a very different pattern can be observed, where the identifiability is 94% for layer 12 and then almost monotonically increases when testing on earlier layers. Further, for $l = 6$ we see both patterns.

This experiment suggests that the nature of token identity changes as tokens pass through the model, and patterns learned on data from later layers transfer well to earlier layers. The experiment also shows that layer 12 is behaving differently than the other layers. In particular, generalizing *to* layer 12 from layer 11 seems to be difficult, signified by a sudden drop in token identifiability rate. We believe this is due to a task dependent parameter adaptation induced in the last layer by the next-sentence prediction task which only uses the CLS token (cf. Appendix B.4 for additional hints that the last layer(s) behave differently). See Appendix B.3 for results of $\hat{g}_{l2,l}^{lin}$, $\hat{g}_{l2,l}^{MLP}$ and $\hat{g}_{cos,l}^{MLP}$.

Overall, the results of this section suggest that one can associate most hidden embeddings with their input token, for example for drawing conclusions based on (effective) attention weights. However, self-attention has the potential to strongly mix tokens across multiple layers, and hence it is unclear whether token identifiability alone is enough to equate hidden embeddings with their input words, or whether we also need to take into account exactly *how much* of the word is still contained in the hidden embedding. In order to address this question, we now study the degree of information mixing among embeddings, and introduce a tool to track the contributions of tokens to embeddings throughout the model.

## 5 ATTRIBUTION ANALYSIS TO IDENTIFY CONTEXT CONTRIBUTION

We consider the role of the contextual information in the hidden embeddings, which is accumulated through multiple paths in a multi-layer network. To shed more light on this process, we introduce *Hidden Token Attribution*, a context quantification method based on gradient attribution (Simonyan et al., 2014) to investigate the hidden tokens' sensitivity with respect to the input tokens.

### 5.1 HIDDEN TOKEN ATTRIBUTION

Gradient based attribution approximates the neural network function $f(\boldsymbol{X})$ around a given sequence of input word embeddings $\boldsymbol{X} \in \mathbb{R}^{d_s \times d}$ by the linear part of the Taylor expansion:

$$f(\boldsymbol{X} + \Delta\boldsymbol{X}) \approx f(\boldsymbol{X}) + \nabla_{\boldsymbol{X}} f(\boldsymbol{X})^T \cdot \Delta\boldsymbol{X} \qquad (10)$$

With this, the network sensitivity is analyzed by looking at how small changes $\Delta\boldsymbol{X}$ at the input correlate with changes at the output. Since in the linear approximation this change is given by the gradient $\nabla_{\boldsymbol{x}_i} f = \frac{\delta f(\boldsymbol{X})}{\delta \boldsymbol{x}_i}$ for a change in the $i$-th input token $\boldsymbol{x}_i \in \mathbb{R}^d$ of $\boldsymbol{X}$, the attribution of how much input token $\boldsymbol{x}_i$ affects the network output $f(\boldsymbol{X})$ can be approximated by the $L_2$ norm

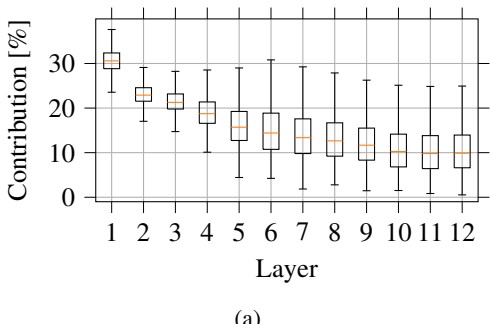 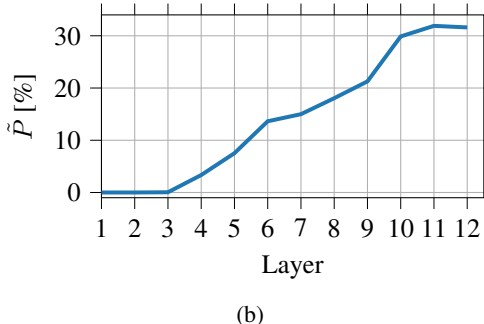

(a)                                             (b)

Figure 3: (a) Contribution of the input token to the embedding at the same position. The orange line represents the median value and outliers are not shown. (b) Percentage of tokens $\tilde{P}$ that are *not* the main contributors to their corresponding contextual embedding at each layer.

of the respective gradient: $\text{attr}(\boldsymbol{x}_i) = ||\nabla_{\boldsymbol{x}_i} f||_2$. Since we are interested in how much a given hidden embedding $\boldsymbol{e}_j^l$ attributes to the input tokens $\boldsymbol{x}_i$, $i \in [1, 2, \ldots, d_s]$, we define the relative input contribution $c_{i,j}^l$ of input $\boldsymbol{x}_i$ to output $f(\boldsymbol{X}) = \boldsymbol{e}_j^l$ as

$$c_{i,j}^l = \frac{||\nabla_{i,j}^l||_2}{\sum_{k=0}^{d_s}||\nabla_{k,j}^l||_2} \qquad \text{with} \quad \nabla_{i,j}^l = \frac{\delta \boldsymbol{e}_j^l}{\delta \boldsymbol{x}_i} \tag{11}$$

Since we normalize by dividing by the sum of the attribution values to all input tokens, we obtain values between 0 and 1 that represent the *contribution* of each input token $\boldsymbol{x}_i$ to the hidden embedding $\boldsymbol{e}_j^l$. *Hidden Token Attribution* differs from the standard use of gradient attribution in that, instead of taking the gradients of the output of the model with respect to the inputs in order to explain the model's decision, we calculate the contribution of the inputs to intermediate embeddings in order to track the mixing of information. Further details of this method are discussed in Appendix C.1.

## 5.2   TOKEN MIXING: CONTRIBUTION OF INPUT TOKENS

We use Hidden Token Attribution to extend the results of Section 4 showing *how much* of the input token is contained in a given hidden embedding. In Figure 3a we report the contribution $c_{j,j}^l$ of input tokens $\boldsymbol{x}_j$ to their corresponding hidden embeddings $\boldsymbol{e}_j^l$ at the same position $j$ for each layer $l$. After the first layer the median contribution of the input token is less than a third (30.6%). The contribution then decreases monotonically with depth; at layer 6 the median is only 14.4% and after the last layer it is 10.7%. In Appendix C.5 we provide detailed results by word type. Next, we study which input token is the largest contributor to a given hidden embedding $\boldsymbol{e}_j^l$. The corresponding input token $\boldsymbol{x}_j$ generally has the largest contribution. Figure 3b shows the percentage $\tilde{P}$ of tokens that are *not* the highest contributor to their hidden embedding at each layer. In the first three layers the original input $\boldsymbol{x}_j$ always contributes the most to the embedding $\boldsymbol{e}_j^l$. In subsequent layers, $\tilde{P}$ increases monotonically, reaching 18% in the sixth layer and 30% in the last two layers.

These results show that, starting from layer three, self-attention strongly mixes the input information by aggregating context into the hidden embeddings. This is in line with the results from Section 4, where we see a decrease in token identifiability rate after layer three. Nevertheless, $\tilde{P}$ is always higher than the token identifiability error at the same layer, indicating that tokens are mixed in a way that often permits recovering token identity even if the contribution of the original token is outweighed by other tokens. This suggests that there is some "identity information" that is preserved through the layers.

The strong mixing of information questions the common assumption that attention distributions can be interpreted as "how much a word attends to another word". However, the fact that tokens remain identifiable despite information mixing opens a number of new interesting questions to be addressed by future research. In particular, this seeming contradiction may be solved by investigating the space in which hidden embeddings operate: is there a relation between semantics and geometric distance for hidden embeddings? Are some embedding dimensions more important than others?

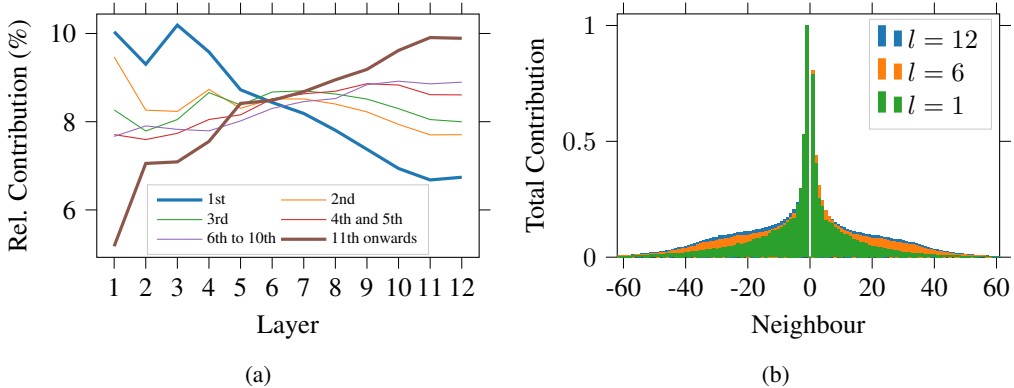

Figure 4: (a) Relative contribution per layer of neighbours at different positions. (b) Total contribution per neighbour for the first, middle and last layers.

### 5.3 Contribution of Context to Hidden Tokens

In this section we study how context is aggregated into hidden embeddings. Figure 4a shows the relative contribution of neighbouring tokens at each layer for the relative positions: first, second, third, fourth and fifth together, sixth to 10th together, and the rest. The closest neighbours (1st) contribute significantly more in the first layers than in later layers. Conversely, the most distant neighbours (11th onwards) contribute the most in deeper layers (cf. Appendix C.2). Despite the progressive increase in long-range dependencies, the context in the hidden embeddings remains mostly local. Figure 4b represents the normalized total contribution aggregated over all tokens from each of their neighbours at the first, middle and last layer. This figure shows that the closest neighbours consistently contribute the most to the contextual word embedding regardless of depth. On the other hand, we indeed observe an increase of distant contributions at later layers.

The results of this section suggest that BERT learns local operators from data in an unsupervised manner, in the absence of any such prior in the architecture. This behavior is not obvious, since attention is a highly non-local operator, and in turn indicates the importance of local dependencies in natural language. While contribution is local *on average*, we find that there are exceptions, such as the [CLS] token (cf. Appendix C.3). Furthermore, using our Hidden Token Attribution method, one can track how context is aggregated for specific tokens (cf. Appendix C.4).

## 6 Related Work

Input-output mappings play a key role in NLP. For example, in machine translation, they were introduced in the form of explicit *alignments* between source and target words (Brown et al., 1993). Neural translation architectures re-introduced this concept in the form of *attention* (Bahdanau et al., 2015). The development of multi-head self-attention (Vaswani et al., 2017) has led to many impressive results in NLP. As a consequence, much work has been devoted to better understand what these models learn, with a particular focus on using attention to explain model decisions.

Jain & Wallace (2019) show that attention distributions of LSTM based encoder-decoder models are not unique, and that adversarial attention distributions that do not change the model's decision can be constructed. They further show that attention distributions only correlate weakly to moderately with dot-product based gradient attribution. Wiegreffe & Pinter (2019) also find that adversarial attention distributions can be easily found, but that these alternative distributions perform worse on a simple diagnostic task. Serrano & Smith (2019) find that zero-ing out attention weights based on gradient attribution changes the output of a multi-class prediction task more quickly than zero-ing out based on attention weights, thus showing that attention is not the best predictor of learned feature importance. Pruthi et al. (2019) demonstrate that self-attention models can be manipulated to produce different attention masks with very little cost in accuracy. These papers differ in their approaches, but they all provide empirical evidence showing that attention distributions are not unique with respect to downstream parts of the model (e.g., output) and hence should be interpreted with

care. Here, we support these empirical findings by presenting a theoretical proof of the identifiability of attention weights. Further, while these works focus on RNN-based language models with a single layer of attention, we instead consider multi-head multi-layer self-attention models. Our token classification and token mixing experiments show that non-identifiable tokens increase with depth, further reinforcing the point that the factors that contribute to the mixing of information are complex and deserve further study.

Voita et al. (2019) and Michel et al. (2019) find that only a small number of heads in BERT have a relevant effect on the output. These results are akin to ours about the non-identifiability of attention weights, showing that a significant part of attention weights do not affect downstream components. One line of work investigates the internal representations of transformers by attaching probing classifiers to different parts of the model. Tenney et al. (2019) find that BERT has learned to perform steps from the classical NLP pipeline. Similarly, Jawahar et al. (2019) show that lower layers of BERT learn syntactic features, while higher layers learn semantic features. They also argue that long-range features are learned in later layers, which agrees with our attribution-based experiments.

## 7 CONCLUSION

We used the notion of identifiability to gain a better understanding of transformers from different yet complementary angles. We started by proving that attention weights are non-identifiable when the sequence length is longer than the attention head dimension. Thus, infinitely many attention distributions can lead to the same internal representation and model output. As an alternative, we propose *effective attention*, a method that improves the interpretability of attention weights by projecting out the null space. Second, we show that tokens remain largely identifiable through a learned linear transformation followed by a nearest neighbor lookup based on cosine similarity. However, input tokens gradually become less identifiable in later layers. Finally, we present *Hidden Token Attribution*, a gradient-based method to quantify information mixing. This method is general and can be used to investigate contextual embeddings in self-attention based models. In this work, we use it to demonstrate that input tokens mix heavily inside transformers. This result means that attention-based interpretations, which suggest that a word at some layer is attending to another word can be improved by accounting for how the tokens are mixed inside the model. We further show that context is progressively aggregated into the hidden embeddings while some identity information is preserved. Moreover, we show that context aggregation is mostly local and that distant dependencies become relevant only in the last layers, which highlights the importance of local information for natural language understanding. Our results suggest that some of the conclusions in prior work (Vaswani et al., 2017; Vig, 2019; Marecek & Rosa, 2018; Clark et al., 2019; Raganato & Tiedemann, 2018; Voita et al., 2019; Tang et al., 2018; Wangperawong, 2018; Padigela et al., 2019; Baan et al., 2019; Lin et al., 2019; Dehghani et al., 2019; Zenkel et al., 2019; Klein & Nabi, 2019; Coenen et al., 2019) may be worth re-examining from this perspective.

There are still many open questions for future research. For one, by constraining *effective attention* to the probability simplex, one could better compare it to standard attention, although in this case non-influencing parts would be included in the weights. More research is needed to better understand the differences between these formulations. Further, we find that tokens mix *and* remain largely identifiable. While these two conclusions are not necessarily at odds - a token can both gather context information and still retain the essence of the original input word - we believe that the relationship between mixing and identifiability warrants further investigation. Moreover, it is becoming increasingly difficult to compare all the new transformer variants, and it is hence important to gain a deeper understanding of this class of models. The concepts introduced in this paper could help in identifying fundamental differences and commonalities between variants of self-attention models.

### ACKNOWLEDGEMENTS

We would like to thank the reviewers and area chairs for their thorough technical comments and valuable suggestions. We would also like to thank Jacob Devlin for feedback on an early draft. This research was supported with Cloud TPUs from Google's TensorFlow Research Cloud (TFRC).

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

## A    IDENTIFIABILITY OF SELF-ATTENTION

### A.1    BACKGROUND ON ATTENTION IDENTIFIABILITY

Often, the identifiability issue arises for a model with a large number of unknown parameters and limited observations. Taking a simple linear model $y = x_1\beta_1 + x_2\beta_2$ as an example, when there is only one observation $(y, x_1, x_2)$, model parameters $\beta_1$ and $\beta_2$ cannot be uniquely determined. Moreover, in the matrix form $Y = X\beta$, by definition the parameter $\beta$ is identifiable only if $Y = X\beta_1$ and $Y = X\beta_2$ imply $\beta_1 = \beta_2$. So if the null space contains only the zero solution $\{\beta|X\beta = 0\} = \{0\}$, i.e., $X\beta_1 - X\beta_2 = X(\beta_1 - \beta_2) = 0 \implies \beta_1 - \beta_2 = 0$, the model is identifiable. Therefore, the identifiability of parameters in a linear model is linked to the dimension of the null space, which in the end is determined by the rank of $X$.

### A.2    ADDITIONAL RESULTS OF THE EFFECTIVE ATTENTION VS. RAW ATTENTION RESULTS

In Figure 5 we provide a recreation of the figure regarding the attention of tokens towards the [SEP] token found in (Clark et al., 2019, Figure 2) with average attention as well as average effective attention. Again, we see that most of the raw attention lies effectively in the null space, confirming the pattern of Figure 1. The figures are produced using the code from Clark et al. (2019).

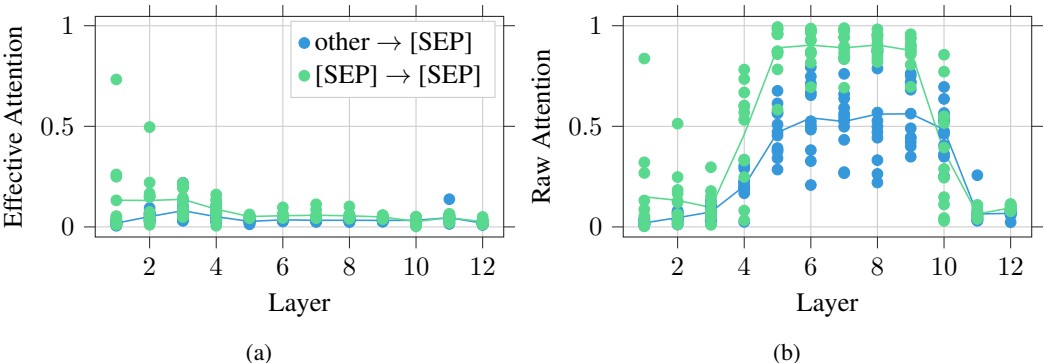

(a)                                   (b)

Figure 5: Effective attention (a) vs. raw attention (b). (a) Each point represents the average effective attention from a token type to a token type. Solid lines are the average effective attention of corresponding points in each layer. (b) is the corresponding figures using raw attention weights.

### A.3    A CLOSER LOOK AT EFFECTIVE ATTENTION WEIGHTS

Here we discuss an example of how effective attention might lead to interpretive conclusions that differ from raw attention. Figure 6 plots the attention weights (raw, effective, null) from one of the attention heads in BERT's layer 4, for the following passage:

"[CLS] research into military brats has consistently shown them to be better behaved than their civilian counterparts. [SEP] hypotheses as to why brats are better behaved: firstly, military parents have a lower threshold for misbehavior in their children; secondly, the mobility of teenagers might make them less likely to attract attention to themselves, as many want to fit in and are less secure with their surroundings; and thirdly, normative constraints are greater, with brats knowing that their behavior is under scrutiny and can affect the military member's career. teenage years are typically a period when people establish independence by taking some risks away from their [SEP]".

For readability, on the y-axis, we consider just the sentence "the mobility of teenagers might make them less likely to attract attention to themselves, as many want to fit in and are less secure with their surroundings".

The following seems worth noticing:

- Raw attention weights are by and large concentrated either on the structural components, [CLS] and [SEP], or on the semi-monotonic, near diagonal alignments.

- Effective attention weights are more uniform, in general. They are still concentrated near the diagonal elements, although less so than in raw attention. However, the attention on [CLS] and [SEP] has vanished. The collapse of the [CLS] and [SEP] weights brings to the surface other interesting things. As an example, we point out the highest weight on the attention matrix, that is not on the diagonal. This involves (highlighted by means of the yellow lines) the main verb of the selected sentence, "make", whose object is "them" (teenagers), and the pronoun "them" (the direct object of the first sentence, "military brats", 48 positions away). The two are co-referential, as both refer to the main subject of the passage, military brats.

- Null attention weights are also more uniform than raw attention ones. Interestingly, they seem to carry all the mass of the [CLS] and [SEP] tokens. There is a visible degree of redundancy between the null attention weights and the effective ones, but also clear complementary elements.

One should not extrapolate too much from a single observation. Further research is needed on this topic. However, this example is a proof of concept that raw and effective attention can diverge qualitatively, in significant ways. It agrees with the hypothesis that the weights on the structural components may act as sinks, as observed in (Clark et al., 2019), but also tells us how this happens. Furthermore, it indicates that attention in the null space can obfuscate other valuable interactions that may be recoverable by inspecting effective attention.

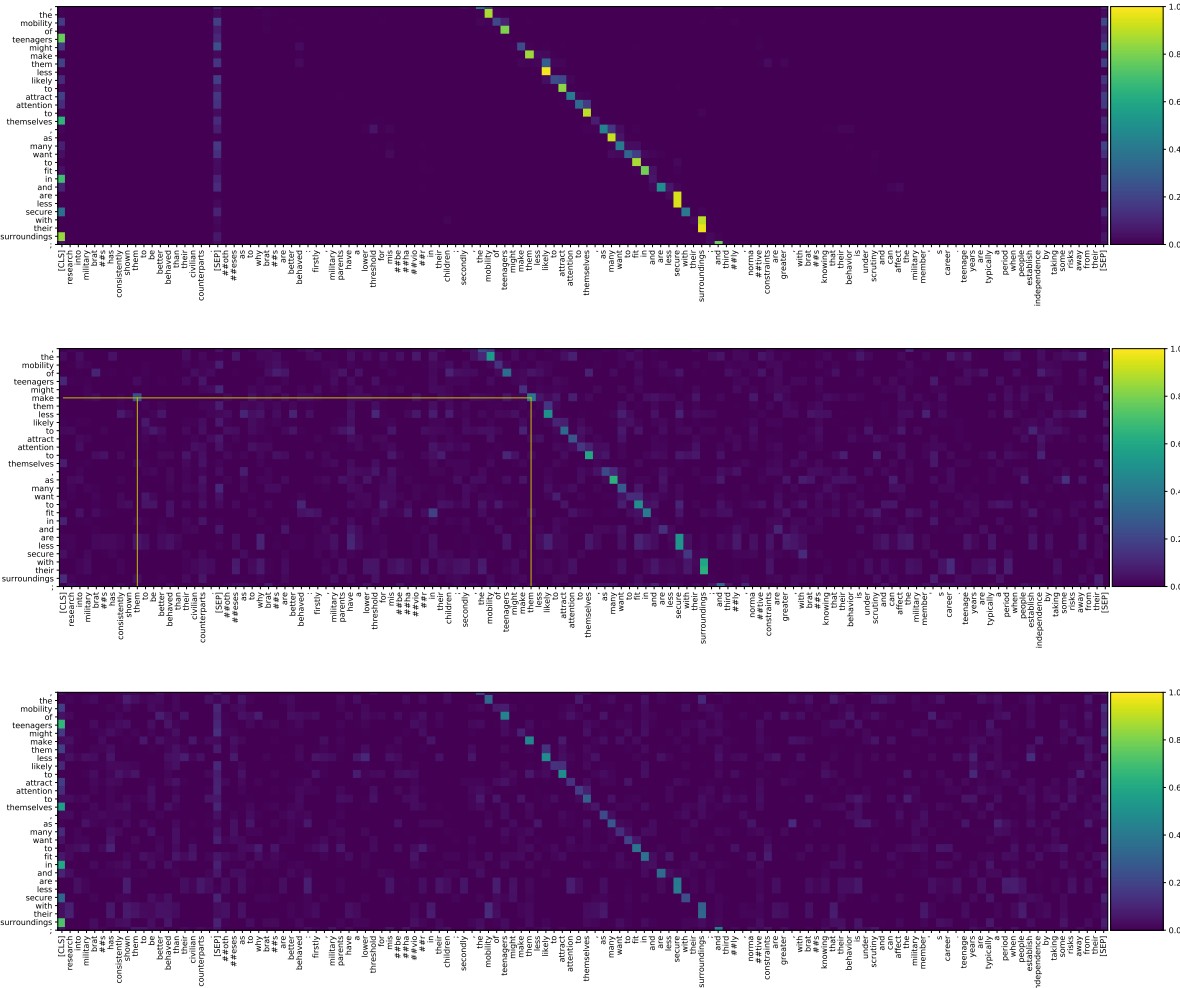

Figure 6: Raw attention weights (top), Effective attention weights (middle) and Null attention weights (bottom).

# B    TOKEN IDENTIFIABILITY EXPERIMENTS

## B.1    EXPERIMENTAL SETUP AND TRAINING DETAILS

The linear perceptron and MLP are both trained by either minimizing the L2 or cosine distance loss using the ADAM optimizer (Kingma & Ba, 2015) with a learning rate of $\alpha = 0.0001$, $\beta_1 = 0.9$ and $\beta_2 = 0.999$. We use a batch size of 256. We monitor performance on the validation set and stop training if there is no improvement for 20 epochs. The input and output dimension of the models is $d = 768$; the dimension of the contextual word embeddings. For both models we performed a learning rate search over the values $\alpha \in [0.003, 0.001, 0.0003, 0.0001, 0.00003, 0.00001, 0.000003]$. The weights are initialized with the Glorot Uniform initializer (Glorot & Bengio, 2010). The MLP has one hidden layer with 1000 neurons and uses the gelu activation function (Hendrycks & Gimpel, 2016), following the feed-forward layers in BERT and GPT. We chose a hidden layer size of 1000 in order to avoid a bottleneck. We experimented with using a larger hidden layer of size 3072 and adding dropout to more closely match the feed-forward layers in BERT. This only resulted in increased training times and we hence deferred from further architecture search.

We split the data by sentences into train/validation/test according to a 70/15/15 split. This way of splitting the data ensures that the models have never seen the test sentences (i.e., contexts) during training. In order to get a more robust estimate of performance we perform the experiments in Figure 2a using 10-fold cross validation. The variance, due to the random assignment of sentences to train/validation/test sets, is small, and hence not shown.

## B.2    GENERALIZATION ERROR

Figure 7 shows the token identifiability rate for train and test set for both models, linear and MLP, when using L2 distance. Both models are overfitting to the same degree. The fact that the linear model has about the same generalization error as the MLP suggests that more training data would not significantly increase performance on the test set. Further, we trained the MLP on layer 11 using 50%, 80%, 90% and 100% of the training data set. The MLP achieved the following token identifiability rate on the test set: 0.74, 0.8, 0.81, 0.82. This indicates that the MLP would not profit much from more data.

We do not report the generalization error for the models trained to minimize cosine distance, as the linear and non-linear perceptrons perform almost equally.

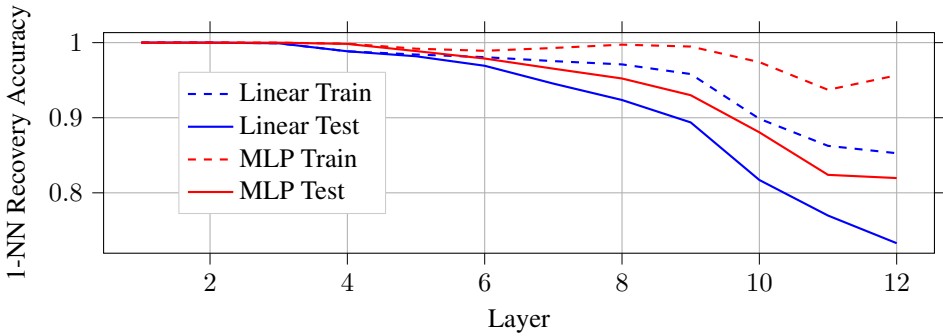

Figure 7: Train and test token identifiability rates for the linear perceptron and MLP.

### B.3 ADDITIONAL RESULTS FOR FIGURE 2B

Figure 2b in the main text only shows results of the linear perceptron trained to minimize cosine distance on layers $l = [1, 6, 11, 12]$ and tested on all other layers. Figures 8, 9 and 10 show the corresponding results for the linear perceptron trained to minimize cosine distance, and for the MLP trained to minimize L2 and cosine distance respectively. Overall, all figures show the same qualitative trends as presented in Section 4 of the main text: Generalizing to later layers works considerably worse than the other way around. The linear perceptrons seem to generalize better across layers, likely due to the MLPs overfitting more to the particular layers they are trained on.

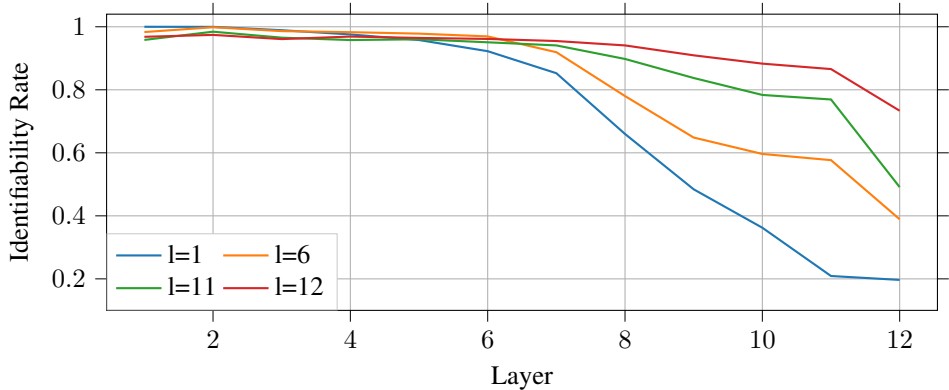

Figure 8: Linear Perceptron trained to minimize L2 distance generalizing to all layers.

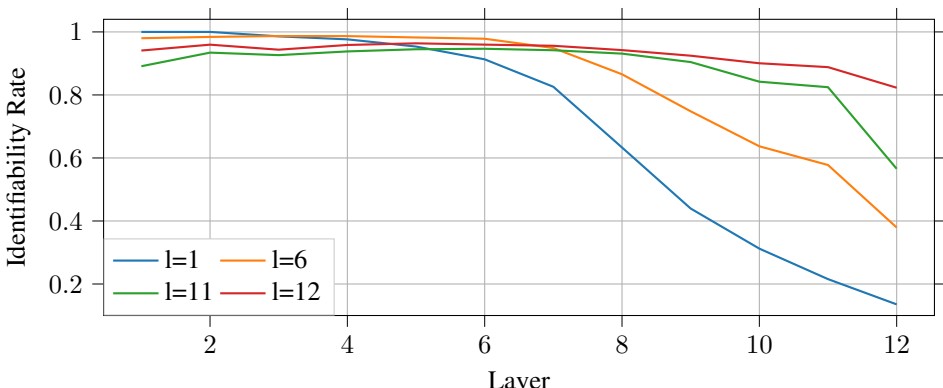

Figure 9: MLP trained to minimize L2 distance generalizing to all layers.

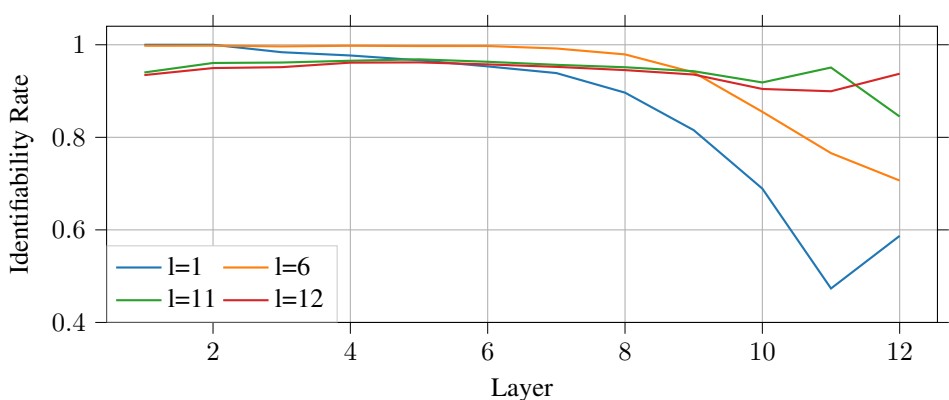

Figure 10: MLP trained to minimize cosine distance generalizing to all layers.

## B.4 TOKEN IDENTITY - FROM HIDDEN TOKENS TO HIDDEN TOKENS

Figure 11 shows results for identifying tokens across single layers of BERT, i.e., the input to $g$ is $(e_i^1, x_i)$ in the first layer, and subsequently $(e_i^l, e_i^{l-1})$, where $l = [1, ..., 12]$. This experiment gives further insight into what kinds of transformations are applied by each transformer layer separately, as opposed to the cumulative transformations shown in Section 4 of the main text. Interestingly, even the naive baselines perform well across single layers. This shows that BERT only applies small changes to the contextual word embeddings, whereas overall the angle (as indicated by the naive baseline using cosine distance) is affected less than the magnitude of the word embeddings (indicated by the naive baseline using L2 distance).

Figure 11 shows that tokens are on average more difficult to identify across later layers. In the main text we hypothesize that the qualitative change seen in later layers could be due to a task-specific parameter adaptation during the second (next sentence prediction) pre-training phase. A possible reason is that during this pre-training-phase, BERT only needs the [CLS] token in the last layer, which is qualitatively very different form the first (masked language modeling) pre-training phase, where potentially all the tokens are needed in the last layer.

To further verify this hypothesis we experimented with BERT fine-tuned on two datasets, MRPC and CoLA (Warstadt et al., 2018). During the fine-tuning phase, similar to the next sentence prediction pre-training phase, only the [CLS] token is needed at the last layer. If task-dependent parameter adaptation indeed has a different influence on the last layer(s) than on earlier layers, then we should be able to see a difference between the finetuned and non-finetuned cases. Figures 12 and 13 compare the naive baselines across single layers for BERT finetuned on MRPC and CoLA, respectively. Indeed, one can see a remarkable decrease in identifiability across the last layer for L2-based nearest neighbour lookup, further indicating that the last layers are indeed more strongly affected by different fine-tuning objectives. Nearest neighbour lookup based on cosine distance is affected much less, indicating that in terms of token identifiability, the last layers are only slightly affected by fine-tuning.

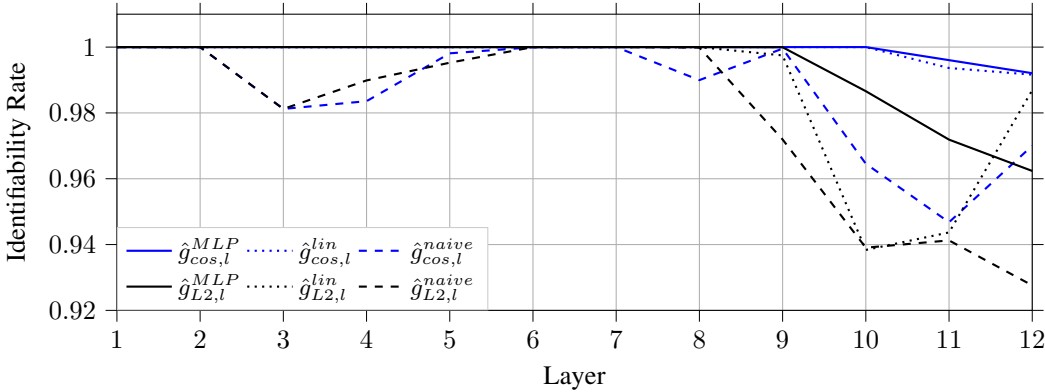

Figure 11: Token identifiability across single layers. These results are for non fine-tuned BERT on MRPC.

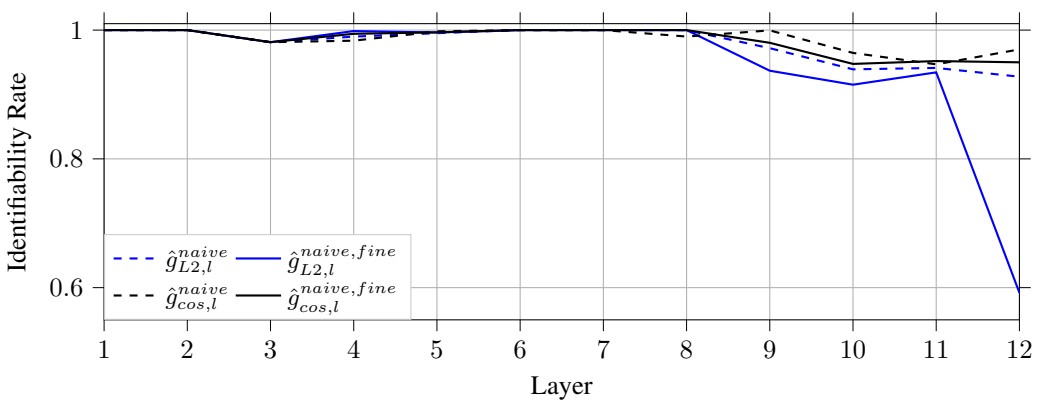

Figure 12: Token identifiability across single layers, comparing non fine-tuned (dashed) BERT against BERT fine-tuned on MRPC (solid).

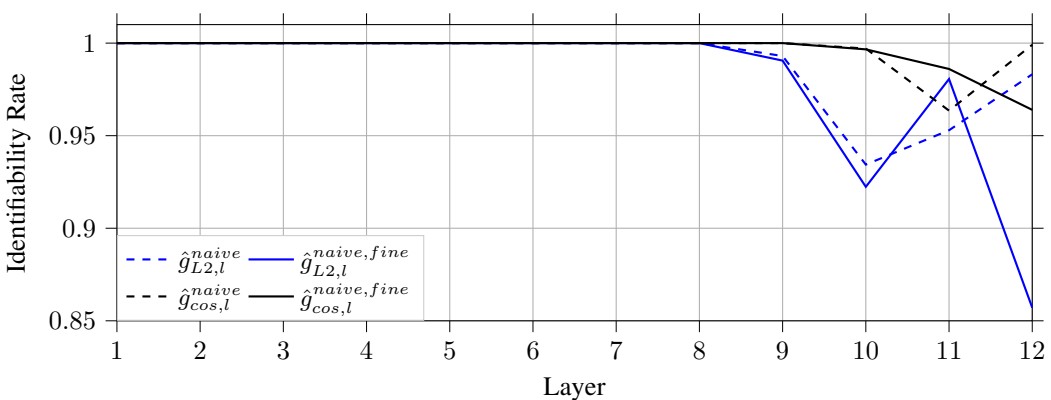

Figure 13: Token identifiability across single layers, comparing non fine-tuned BERT (dashed) against BERT fine-tuned on CoLA (solid).

### B.5 TOKEN IDENTITY - RECOVER NEIGHBOURING INPUT TOKENS

In Section 4 of the main text we show that tokens at position $i$ remain largely identifiable throughout the layers of BERT. In this section we show results of a related experiment, where we test how much information about tokens at neighbouring positions is contained in a contextual word embedding. More formally, the input to $g$ is $(\boldsymbol{e}_i^l, \boldsymbol{x}_{i\pm k})$, where $k \in \{1, 2, 3\}$. Thus, we try to recover input token $\boldsymbol{x}_{i+k}$ from hidden token $\boldsymbol{e}_i^l$. Figures 14, 15, 16 and 17 show the results of for $\hat{g}_{cos,l}^{lin}$, $\hat{g}_{cos,l}^{MLP}$, $\hat{g}_{L2,l}^{MLP}$ and $\hat{g}_{L2,l}^{lin}$, respectively. In the figures, blue corresponds to "previous" tokens and red to "next" tokens.

From the figures we can see that tokens do contain information about neighbouring tokens that lets us recover the neighbouring tokens based on a transformation and subsequent nearest neighbour lookup. The identifiability rate drops both with increasing $k$, but also with increasing depth. Interestingly, recovering left (blue) and right (red) neighbours shows different behaviour, indicating that BERT is treating left and right context differently, despite having been pre-trained using a bi-directional language modeling objective.

Overall, neighbouring tokens can be recovered to a much lower degree than same-position tokens (cf. Section4).

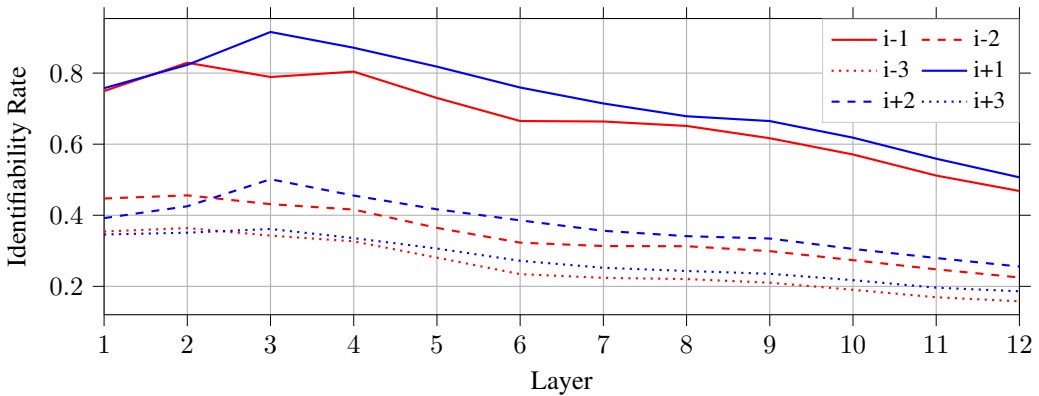

Figure 14: Recovering neighbouring input tokens using $\hat{g}_{cos,l}^{lin}$.

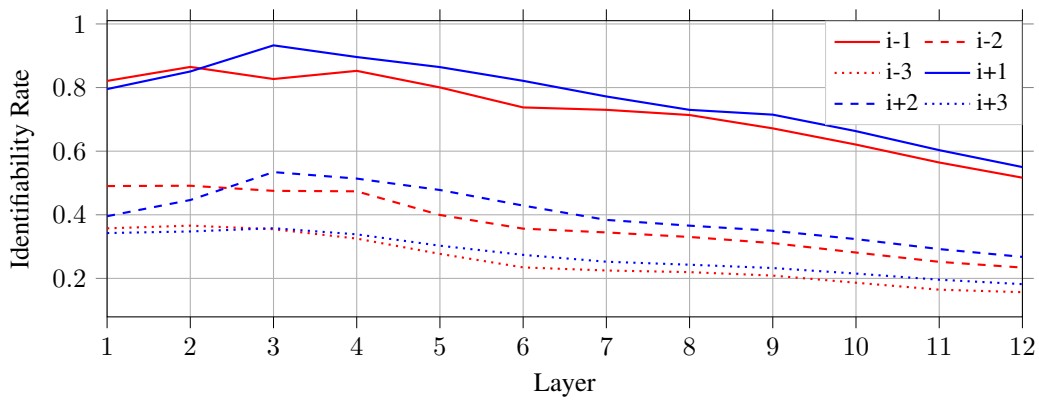

Figure 15: Recovering neighbouring input tokens using $\hat{g}_{cos,l}^{mlp}$.

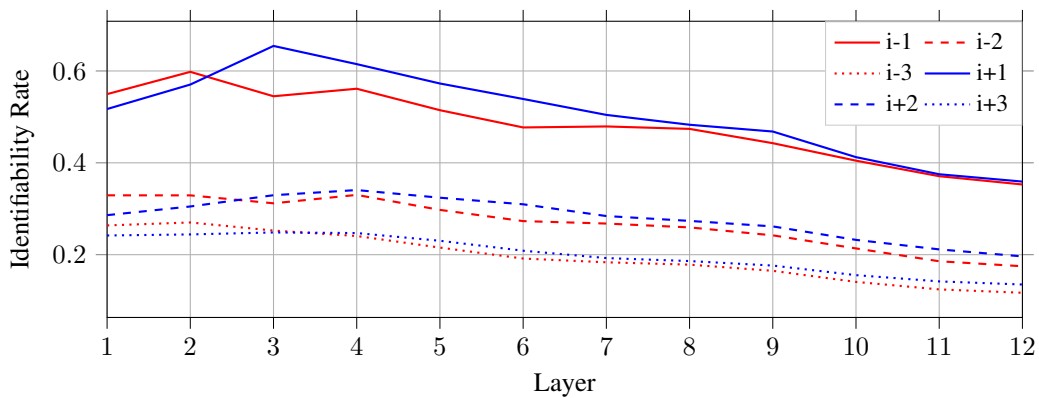

Figure 16: Recovering neighbouring input tokens using $\hat{g}_{L2,l}^{mlp}$.

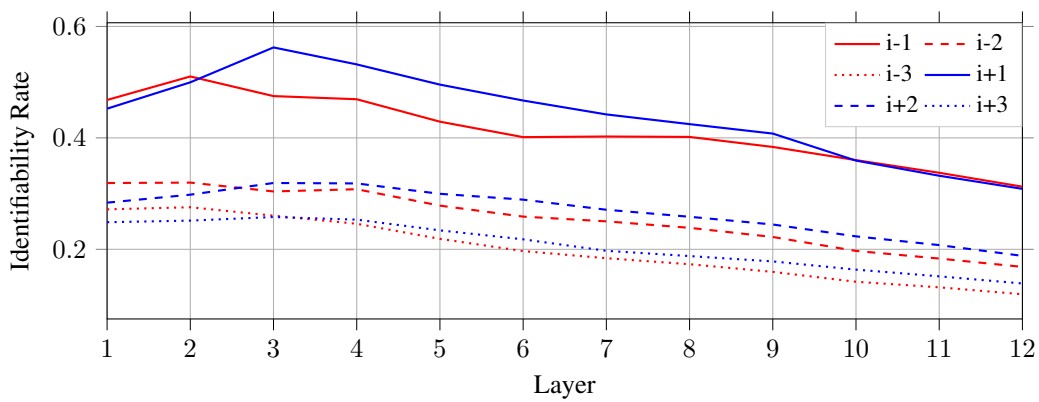

Figure 17: Recovering neighbouring input tokens using $\hat{g}_{l2,l}^{lin}$.

## C   CONTEXT CONTRIBUTION ANALYSIS

### C.1   HIDDEN TOKEN ATTRIBUTION: DETAILS

The attribution method proposed in Section 5.1 to calculate the contribution of input tokens to a given embedding does not look at the output of the model but at the intermediate hidden representations and therefore is task independent. Since the contribution values do not depend on the task that is evaluated, we can compare these values directly to attention distributions, which are also task-independent. In this way, we can compare to other works in the literature (Vig, 2019; Clark et al., 2019; Klein & Nabi, 2019; Coenen et al., 2019; Lin et al., 2019) by using the publicly available pretrained BERT model in our analyses without fine-tuning it to a specific task.

Furthermore, since we are not interested in analysing how the input affects the output of the model but in quantifying the absolute contribution of the input tokens to the hidden embeddings, we use the $L_2$ norm of the gradients. If we were analyzing whether the input contributed positively or negatively to a given decision, the dot-product of the input token embedding with the gradient would be the natural attribution choice (Pörner et al., 2018).

### C.2   CONTEXT IDENTIFIABILITY: DETAILS

To calculate the relative contribution values shown in Figure 4a we firstly calculate the mean of the left and right neighbours for each of the groups of neighbours, i.e., first, second, third, fourth and fifth, sixth to 10th and, from 11th onwards. Then we aggregate the values averaging over all the tokens in the MRPC evaluation set. Finally, we normalize for each group so that the sum of the contribution values of each group is one. In this way, we can observe in which layer the contribution of a given group of neighbours is the largest.

Our results on context identifiability from Section 5.3 complement some of the studies in previous literature. In (Jawahar et al., 2019) the authors observe that transformers learn local syntactic tasks in the first layers and long range semantic tasks in the last layers. We explain this behavior from the point of view of context aggregation by showing that distant context acquires more importance in the last layers (semantic tasks) while the first layers aggregate local context (syntactic tasks). Furthermore, the results showing that the context aggregation is mainly local, specially in the first layers, provide an explanation for the increase in performance observed in (Yang et al., 2018). In that work, the authors enforce a locality constraint in the first layers of transformers, which pushes the model towards the local operators that it naturally tends to learn, as we show in Figure 4b, improving in this way the overall performance.

### C.3   CONTEXT CONTRIBUTION TO CLS TOKEN

In this section we use Hidden Token Attribution to look at the contribution of the context to the [CLS] token, which is added to the beginning of the input sequence by the BERT pre-processing pipeline. This is an especially interesting token to look at because the decision of BERT for a classification task is based on the output in the [CLS] token. Furthermore, like the [SEP] token, it does not correspond to a natural language word and its position in the input sequence does not have any meaning. Therefore, the conclusion that context is on average predominantly local (cf. Section 5.3), is likely to not hold for [CLS].

The second and final pre-training task that BERT is trained on is next sentence prediction. During this task, BERT receives two sentences separated by a [SEP] token as input, and then has to predict whether the second sentence follows the first sentence or not. Therefore, it is expected that the context aggregated into the [CLS] token comes mainly from the tokens around the first [SEP] token, which marks the border between the first and second sentence in the input sequence. In Figure 18 we show the contribution to the [CLS] token from all of its neighbours averaged over all the examples in the MRPC evaluation set for the first, middle and last layers. In Figure 18a, the [CLS] token is placed at position 0 and we see how the context contribution comes mainly from the tokens around position 30, which is roughly the middle of the input examples. In Figure 18b we center the contribution around the first [SEP] token and indeed, it becomes clear that the [CLS] token is aggregating most of its context from the tokens around [SEP], i.e., from the junction between both

sentences. In particular, the two tokens with the highest contribution are the tokens directly before and after [SEP]. Also, it seems that the second sentence contributes more to [CLS] than the first one.

These results give an insight on what information BERT uses to solve next sentence prediction and serves as an illustrative example of how Hidden Token Attribution can be used to analyze transformers.

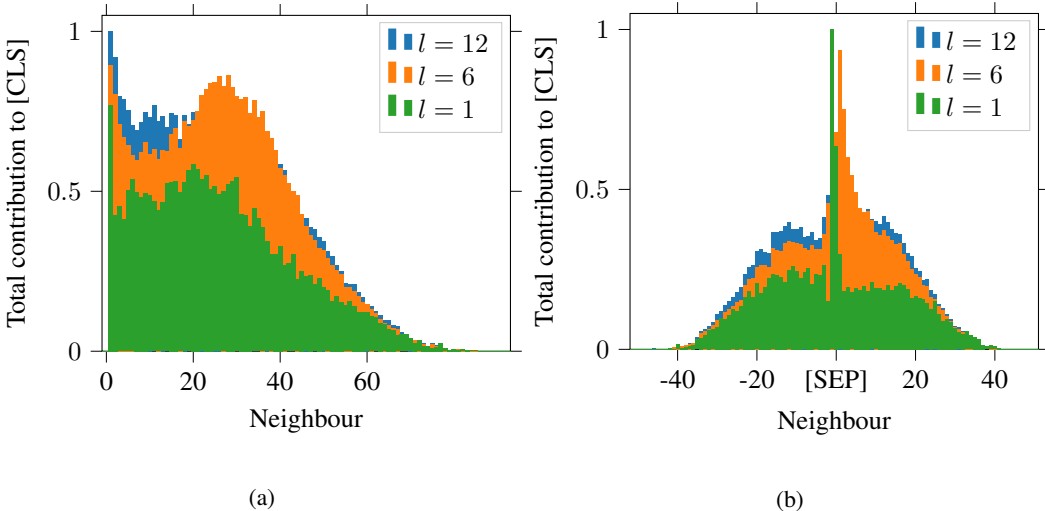

(a)                                                          (b)

Figure 18: Normalized total contribution to the [CLS] token (a) centered around [CLS] at position 0 (b) centered around [SEP].

### C.4 Tracking Context Contribution

Here we show examples of how Hidden Token Attribution can track how context is aggregated for a given word at each layer. For reasons of space we show only few words of a randomly picked sentence of the MRPC evaluation set, which is tokenized as follows:

```
[CLS] he said the foods ##er ##vic ##e pie business doesn ' t fit
the company ' s long - term growth strategy . [SEP] " the foods
##er ##vic ##e pie business does not fit our long - term growth
strategy . [SEP]
```

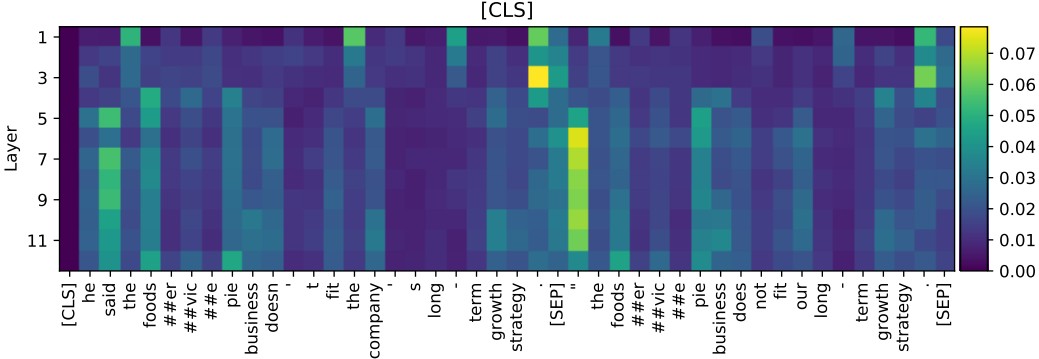

Figure 19: [CLS]: Aggregates context from all tokens but more strongly from those around the first [SEP] token. We hypothesize that this is due to the Next Sentence Prediction pre-training.

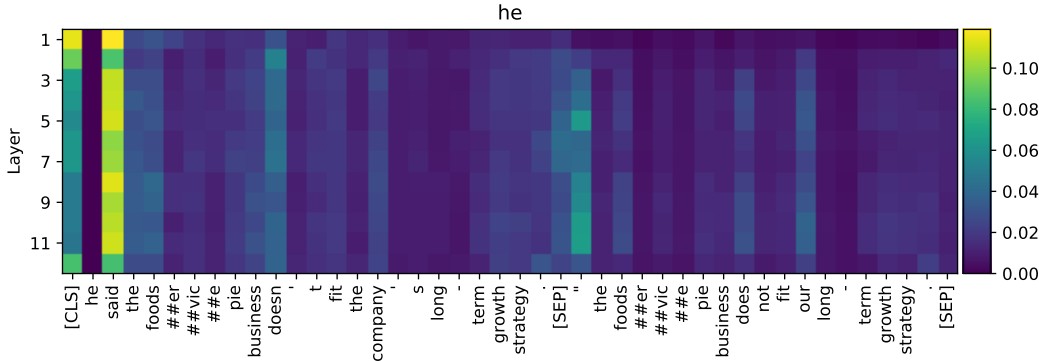

Figure 20: he: Aggregates most context from the main verb of the sentence, "said".

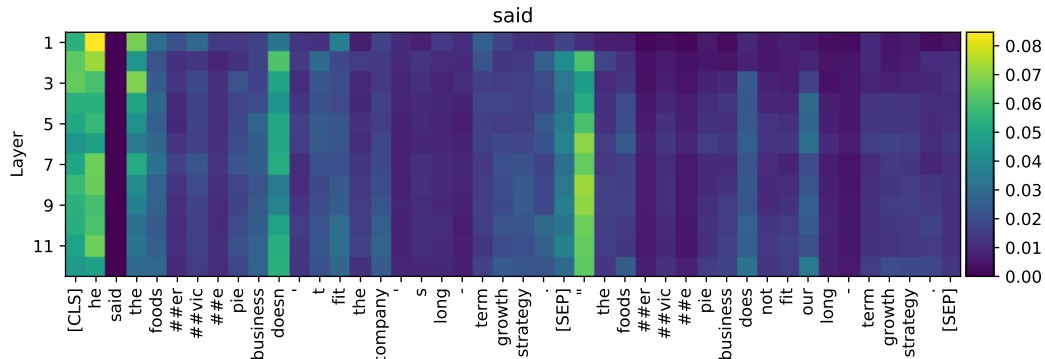

Figure 21: said: Aggregates context mainly from its neighborhood, the main verb of the subordinate sentence and the border between the two input sentences.

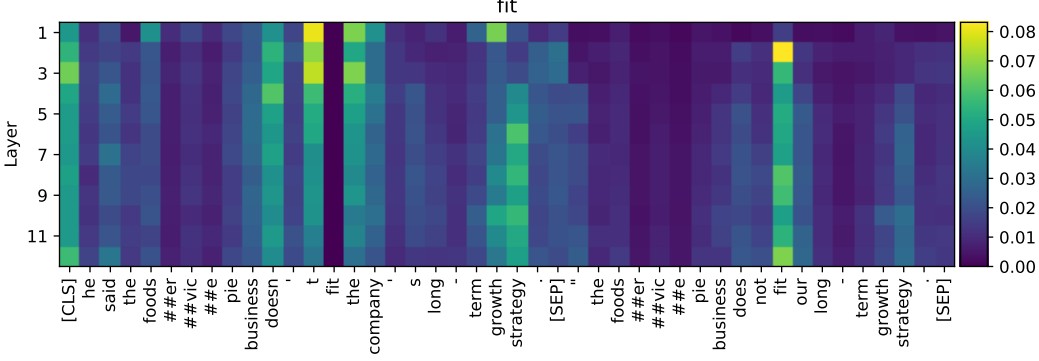

Figure 22: fit: In the first layers it aggregates most context from its neighborhood and towards the last layers it gets the context from its direct object (strategy) and from the token with the same meaning in the second sentence.

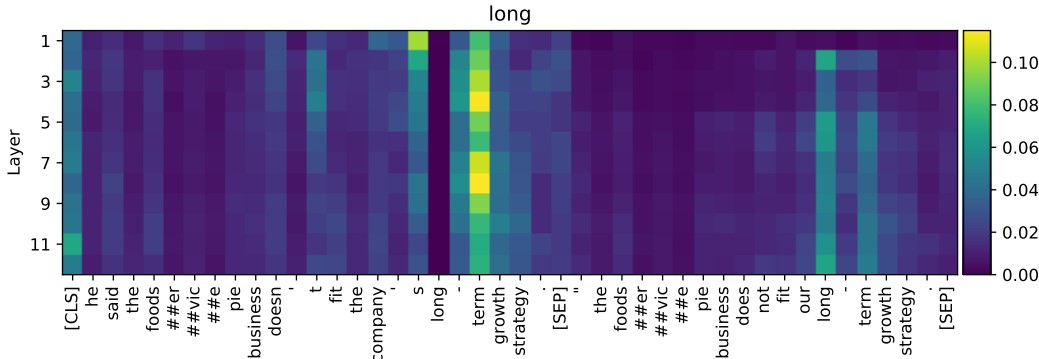

Figure 23: long: It is part of a composed adjective (long-term) and aggregates most of its context from the other part of the adjective (term) as well as from the same tokens in the second sentence. Interestingly, it mostly ignores the hyphen.

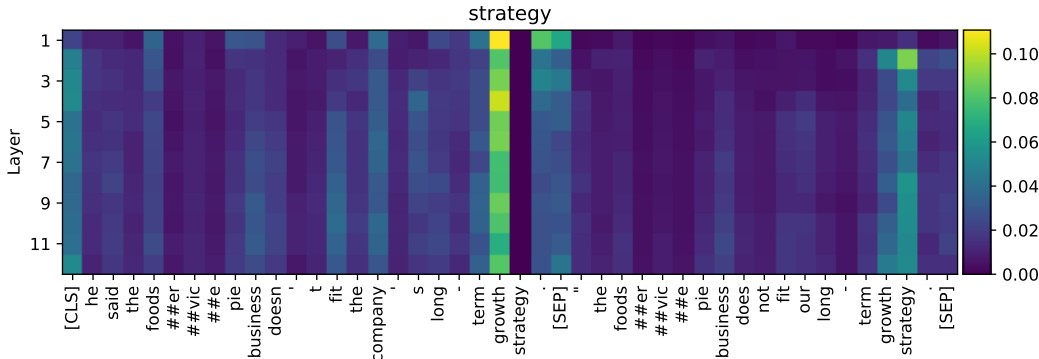

Figure 24: strategy: Aggregates context from the word growth, which is the first one of the noun phrase "growth strategy".

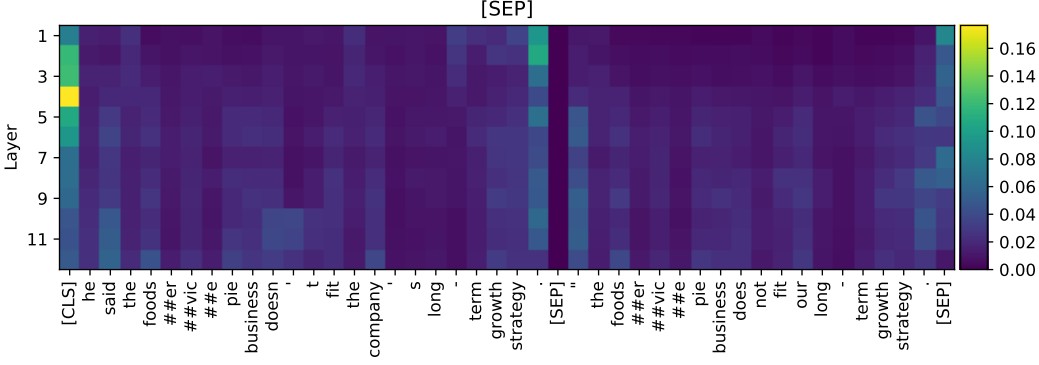

Figure 25: [SEP]: This token that has no semantic meaning aggregates context mostly from [CLS] and its own neighborhood.

## C.5 TOKEN CONTRIBUTIONS BY POS TAG

Here we show the contribution of input tokens to hidden representations in all layers split by part-of-speech (POS) tag (Toutanova et al., 2003). The POS tags are ordered according to the contribution in layer 12.

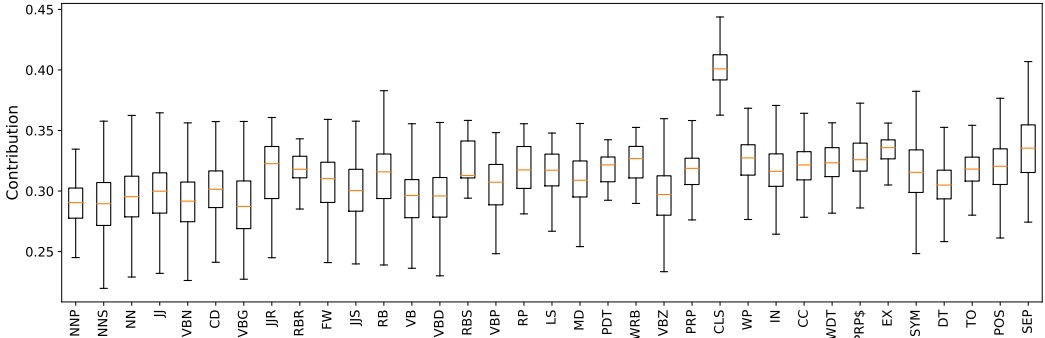

Figure 26: Layer 1: Most token types are equally mixed and have already less than 35% median contribution from their corresponding input. The only exception are the [CLS] tokens, which remain with over 40% median original contribution.

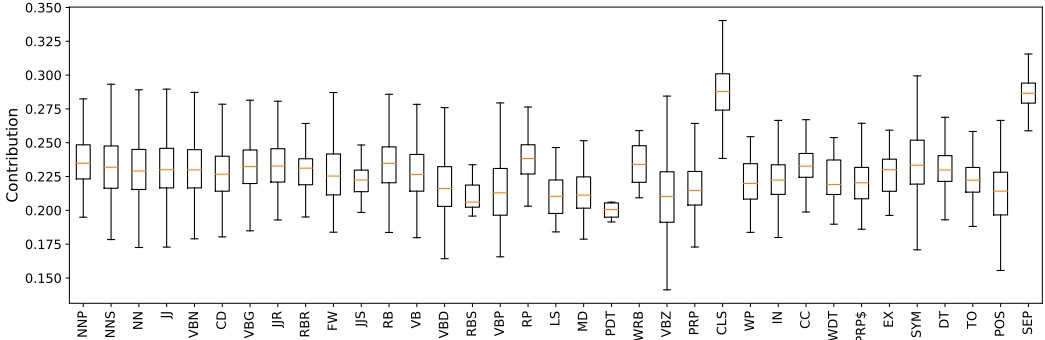

Figure 27: Layer 2: Similar to the previous layer with less contribution over all and [SEP] behaving similarly to [CLS].

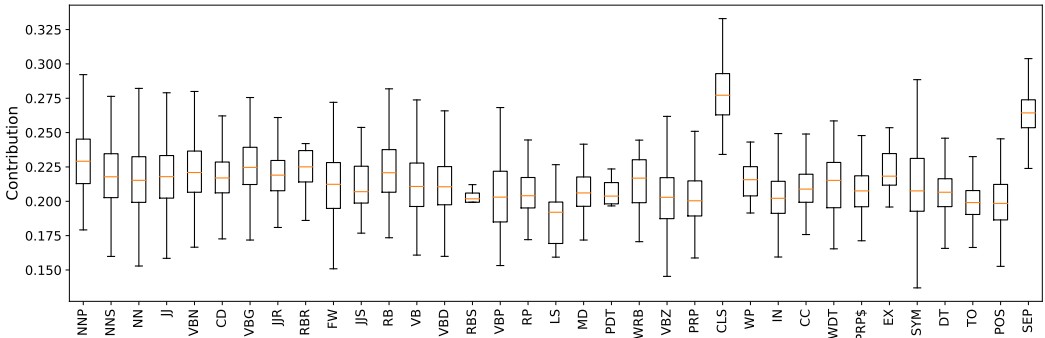

Figure 28: Layer 3: Similar to layer 2 with decreasing contribution overall.

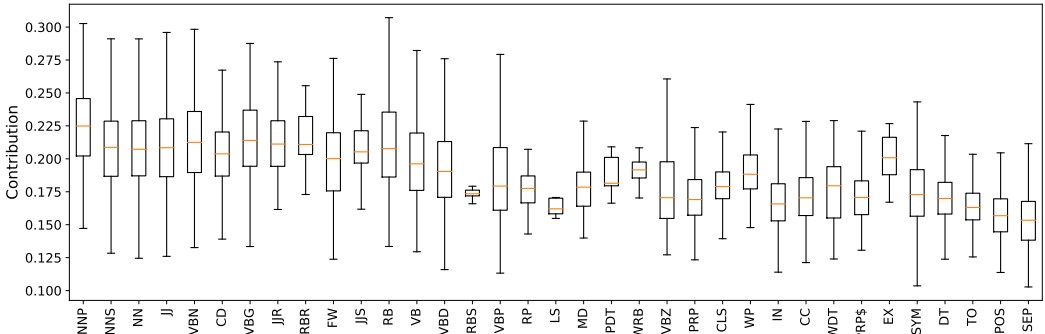

Figure 29: Layer 4: The original input contribution to [CLS] and [SEP] falls significantly. The trend that the word types will follow until the last layer is already clear: Most nouns (NNP, NNS, NN), verbs (VBN, VB, VBD, VBP), adjectives (JJ, JJS) and adverbs (RBR, RBS) keep more contribution from their corresponding input embeddings than words with "less" semantic meaning like Wh-pronouns and determiners (WP, WDT), prepositons (IN), coordinating conjunctions (CC), symbols (SYM), possessives (PRP$, POS) or determiners (DT).

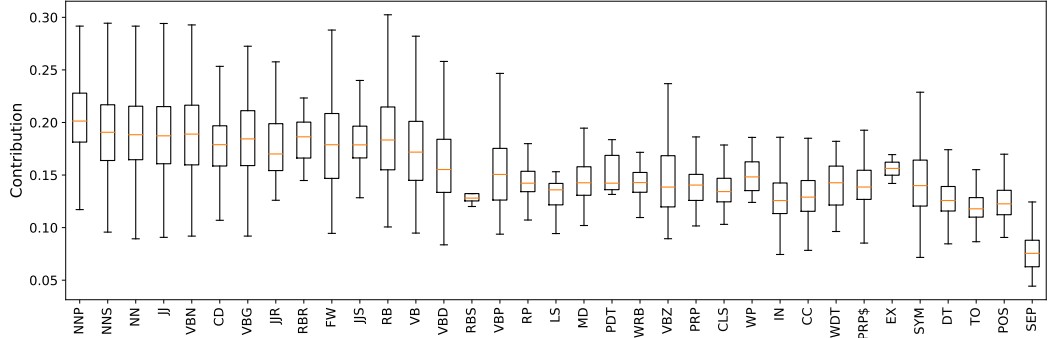

Figure 30: Layer 5: The trend started in the previous layer continues, with a reduction of internal variability within those word types with less original contribution.

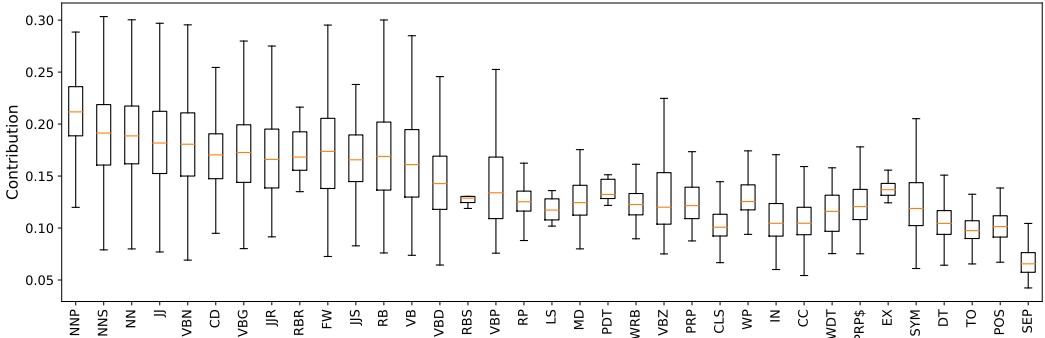

Figure 31: Layer 6: Similar behavior as in the previous layer with minor evolution.

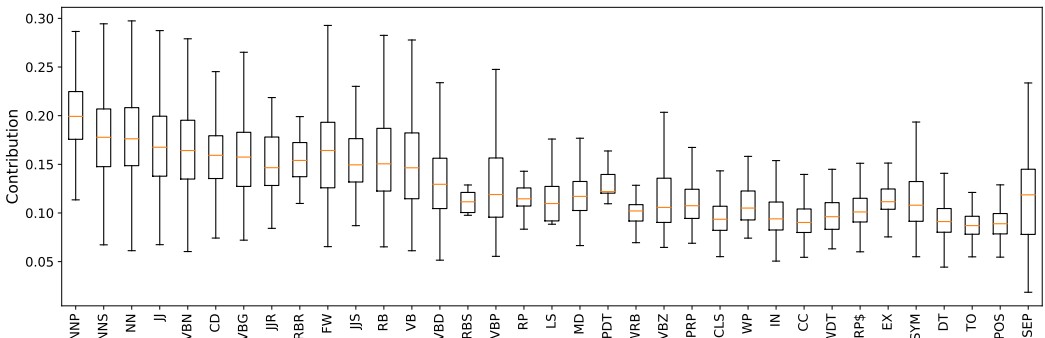

Figure 32: Layer 7: Minor changes with respect to Layer 6.

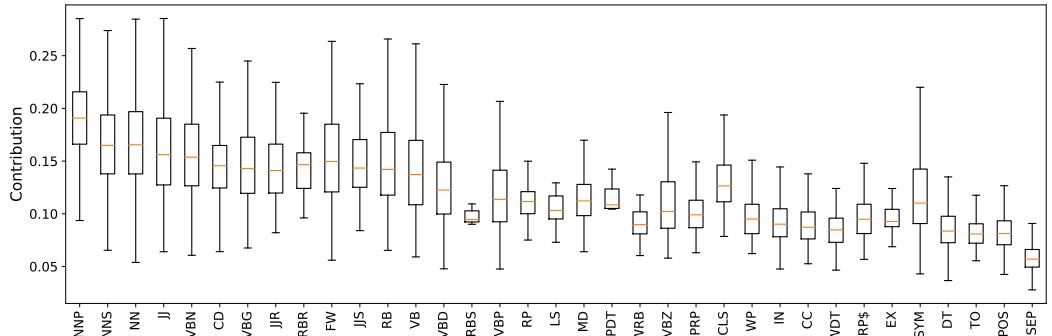

Figure 33: Layer 8: At this point there is clearly a different behavior between the tokens with most contribution which present more intra-class variability, and those with less contribution, which are more uniform.

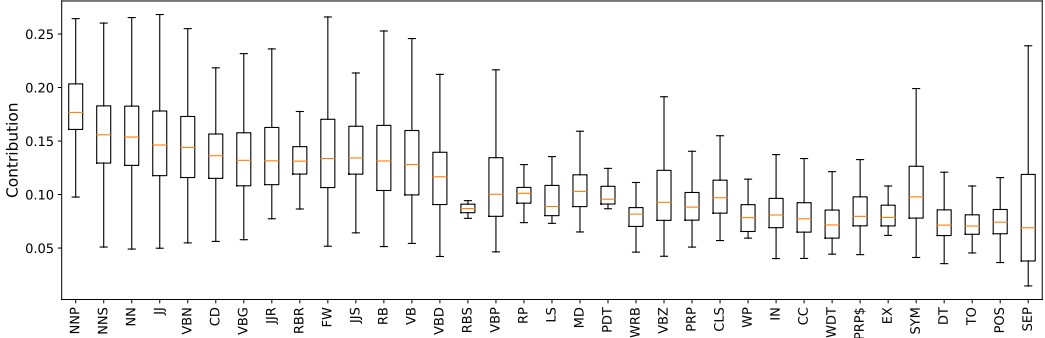

Figure 34: Layer 9: SEP changes increasing the contribution, while the rest stays similar.

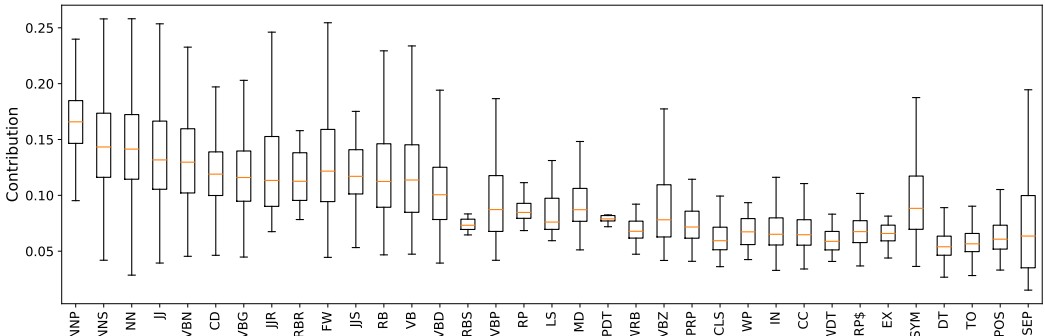

Figure 35: Layer 10: The contribution evolves with the same pattern as in previous layers.

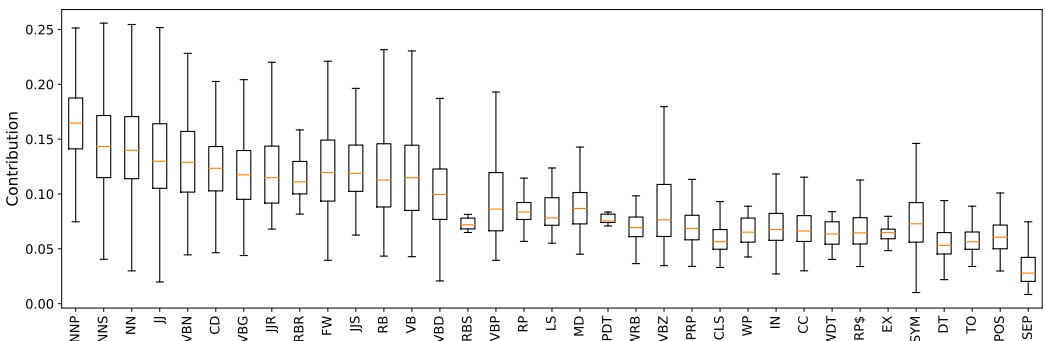

Figure 36: Layer 11:The contribution evolves with the same pattern as in previous layers.

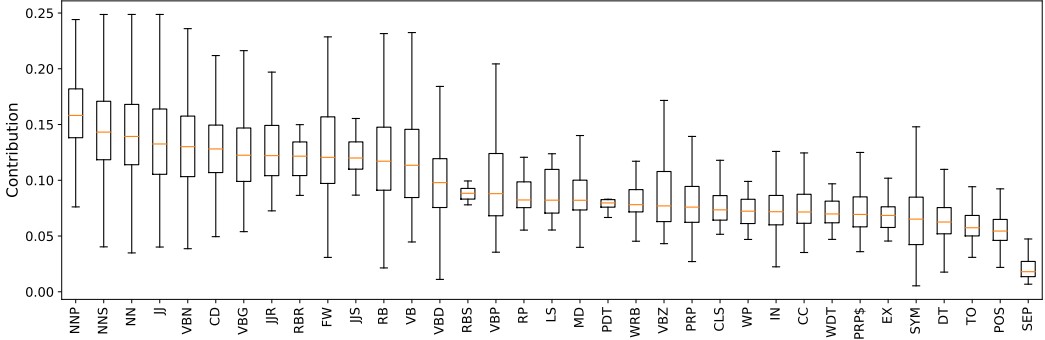

Figure 37: Layer 12: Finally, nouns, verbs, adjectives, adverbs, receive more contribution from their corresponding input than determiners, prepositions, pronouns, "to" words and symbols.

## D  GENERALIZATION TO OTHER DATASETS

In this appendix we reproduce several experiments from the main text using the development sets of two additional datasets from the GLUE benchmark: The Corpus of Linguistic Acceptability (CoLA) (Warstadt et al., 2018), and the matched Multi-Genre Natural Language Inference corpus (MNLI-matched) (Williams et al., 2018). CoLA is a dataset about grammatical acceptability of sentences and MNLI consists of pairs of sentences where the second sentence entails, contradicts or is neutral about the first one. These datasets differ significantly from MRPC. The development set of CoLa has 1043 examples with sequence length $d_s$ between 5 and 35 tokens, and 11 tokens on average. The development set of MNLI-m consits of 9815 examples although we restrict the experiments to the first 4000 examples without loss of generality. These contain a total of 155964 tokens, with a sequence length comprised between 6 and 128 tokens and an average of 39 tokens per example.

The results presented in this appendix are qualitatively similar to those presented in the main text, which shows that our empirical conclusions about BERT are general across data domains.

### D.1  TOKEN IDENTIFIABILITY

Here we reproduce the main token identifiability results of Section 4 on two additional datsets: CoLA and MNLI. Qualitatively, the results are in line with those for MRPC. Note that a random classifier would achieve an accuracy of $1/\bar{d}_s$, where $\bar{d}_s$ denotes the average sentence length. Thus, the random guessing baselines for MRPC, CoLA and MNLI are 1.7%, 9% and 2.6% respectively.

### D.1.1  COLA

Figure 38 shows the token identifiability results for CoLA.

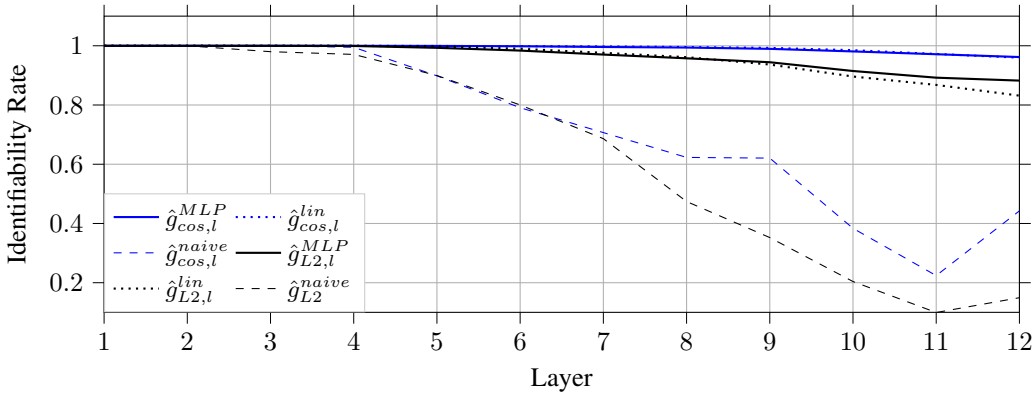

Figure 38: Identifiability of contextual word embeddings at different layers on CoLA.

### D.1.2   MNLI

Figure 39 shows the token identifiability results for the first 500 sentences (19,839 tokens) of MNLI-matched.

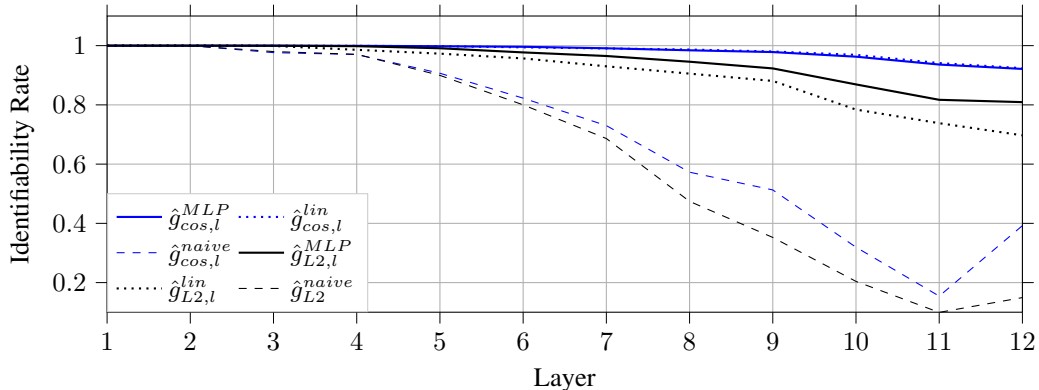

Figure 39: Identifiability of contextual word embeddings at different layers on a the first 500 sentences of MNLI-matched (19,839 tokens).

## D.2 ATTRIBUTION ANALYSIS

### D.2.1 COLA EXPERIMENTS

Figure D.2.1 shows the token mixing analysis for the CoLA dataset. The behavior is very similar to MRPC with the only difference that both, the contribution of the original token and the percentage of tokens that are not maximum contributors to their embeddings are slightly larger across layers. However, this increase is explained by CoLA consisting of much shorter sequences on average than MRPC.

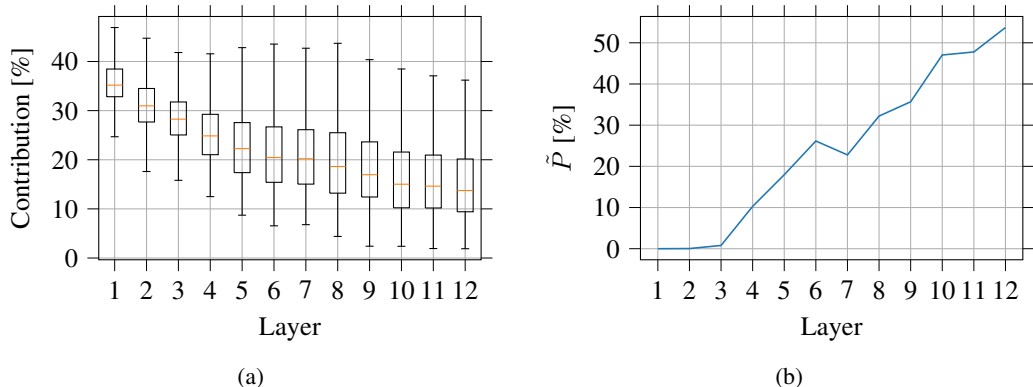

       (a)                                        (b)

Figure 40: (a) Contribution of the input token to the embedding at the same position. (b) Percentage of tokens $\tilde{P}$ that are *not* the main contributors to their corresponding contextual embedding at each layer.

Figure D.2.1 presents the context aggregation for the CoLA development set. We observe the same general trend as for MRPC, with the context being aggregated mostly locally and long range dependencies increasing in the later layers. The fact that examples in CoLA have an average sequence length of 11 tokens explains the smaller relative contribution of tokens beyond the 10th neighbour.

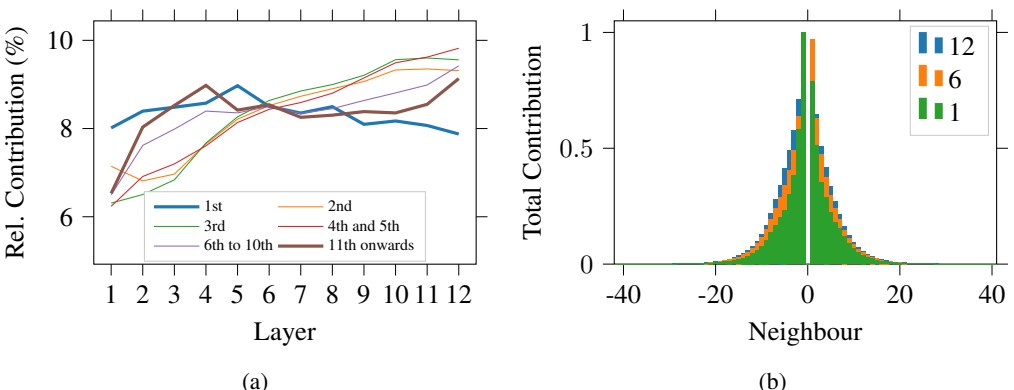

       (a)                                        (b)

Figure 41: (a) Relative contribution per layer of neighbours at different positions. (b) Total contribution per neighbour for the first, middle and last layers.

### D.2.2 MNLI EXPERIMENTS

As shown by Figures D.2.2 and D.2.2, the results with the MNLI matched dataset are very similar to the ones presented in the main text. No meaningful discrepancy exists in this case.

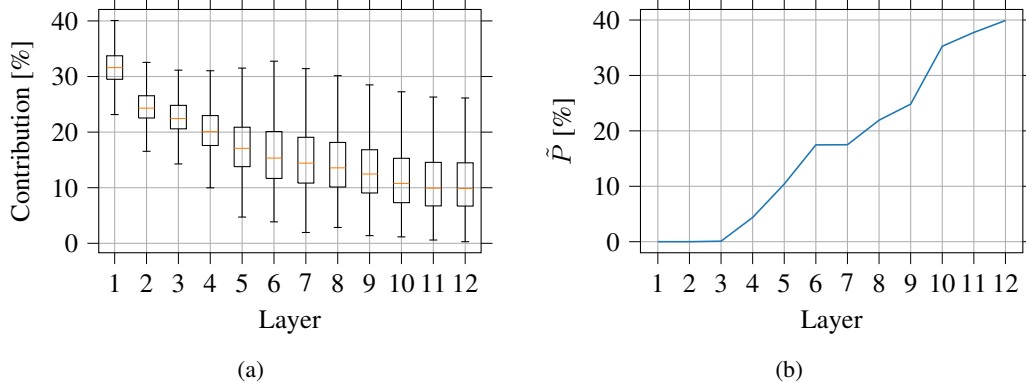

(a)                                                            (b)

Figure 42: (a) Contribution of the input token to the embedding at the same position. (b) Percentage of tokens $\tilde{P}$ that are *not* the main contributors to their corresponding contextual embedding at each layer.

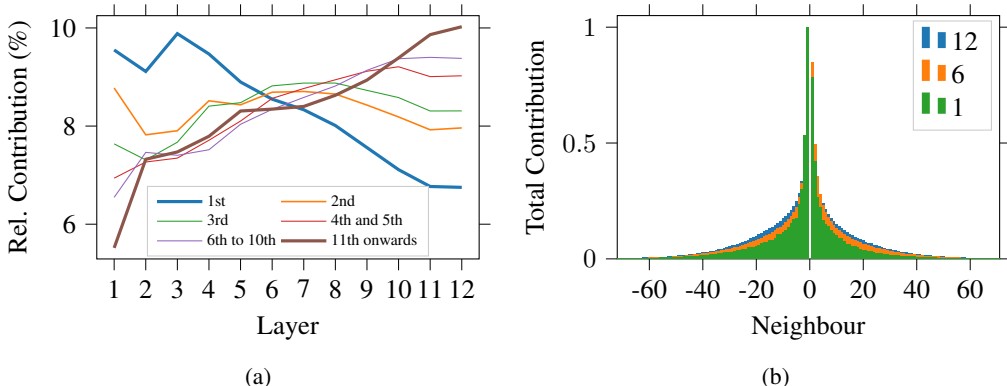

(a)                                                            (b)

Figure 43: (a) Relative contribution per layer of neighbours at different positions. (b) Total contribution per neighbour for the first, middle and last layers.

