# OpenReview forum: "On Identifiability in Transformers"
_ICLR.cc/2020/Conference — Accept (Poster)_

### Official Review · AnonReviewer2 · 2019-10-23
**Official Blind Review #2**

**Rating:** 6

**Review:**

This very interesting paper proposes new perspectives for investigating the self-attention distribution and contextual embedding and challenges many existing findings in the literature. The conclusion presented in this paper partially aligned with some concurrent works (Danish Pruthi et al. 2019, Serrano et al. 2019 and etc.)
Many concrete findings in the paper make sense very much (not surprising), like 1) Input tokens retain their identity in the first a few hidden layers and then progressively become less identifiable, 2) Non-linear activations are crucial in preserving token identity and 3) Strong mixing of input information in the generation of contextual embedding.

The neat proof in the paper, though is based on some assumptions, formally shows that the multi-head self-attention is not always identifiable. The conclusion is that when the sequence length is larger than the size of attention head, there is redundant parameterization space for the attention.
The authors propose “effective attention” which is orthogonal to the null space as a diagnostic tool. In Figure one, the fact that the Pearson correlation between raw and effective attention weight decreases as the sequence length increases experimentally support the theorem.

However, I am not sure what benefit identifiable attention would bring us. It seems that "redundancy" sometimes have some "benefits". As shown in the original Transformer paper, very large d_v or very small d_v both lead to worse performance. I am not sure how “effective attention” would ease the training, though I agree it is a good "diagnostic tool".

Another concern is about how the “non-identifiability” really hurts the model? A pre-trained transformer-based model can achieve very good performance for discriminative tasks after fine-tuning. This shows that a deep model, though there are many uncertain choices in the earlier and middle layers, they finally give us correct discriminative labels.
As the authors claim that contextual embeddings are strong mixing of input (which I agree), my question is "will the non-identifiable attention weights, dramatically affect the mixing component of the contextual embeddings"? Or, will the uncertain choices in earlier layers finally come to close contextual representations in layers closer to output?

Towards the claim that “non-linear activations in preserving token identity”, the authors provide evidence in Figure 2.
Can we also do experiments that using the embedding e^{l}_j (Layer l and position j), through non-linear activations to recover neighboring tokens of position j? I suspect that neighboring tokens can probably also being covered very well by MLP g^{MLP}_l.



**Experience Assessment:**

I have read many papers in this area.

**Review Assessment: Checking Correctness Of Derivations And Theory:**

I carefully checked the derivations and theory.

**Review Assessment: Checking Correctness Of Experiments:**

I carefully checked the experiments.

**Review Assessment: Thoroughness In Paper Reading:**

I read the paper thoroughly.

---

> ### Author Response · Authors · 2019-11-11
> **Response to Reviewer #2**
>
> Thank you very much for the thorough and constructive feedback. Below, we give an answer to the concerns raised in this review.
>
> [The conclusion presented in this paper partially...] Thanks for pointing us towards relevant related work. We already cite Serrano et al., and we have added Pruthi et al. to the related work section.
>
> [However, I am not sure what benefit …] With respect to redundancy, we expect there to be redundancy. We discuss one case (Appendix A.3) that seems to support clearly that there is significant redundancy between raw and null attention, but also clear complementary elements. For example, the attention weights on the structural tokens ([CLS], and [SEP]) vanish in the effective attention. This, in turn, makes it possible that other interesting interactions encoded in the attention weights will become more prominent.
>
> [...I am not sure how “effective attention” would ease the training...] The study of attention identifiability focuses on interpreting a model’s behavior and it does not imply that resolving the unidentifiability of attention would ease the training. In fact, in current transformer architectures, the unidentifiability is always present as long as d_s>d_v.
>
> [Another concern is about how the “non-identifiability” really hurts the model?] The non-identifiability of attention weights does not affect the model in any way, since the output of the self-attention operation is the same in either case. The added value of effective attention comes from the insights that can be derived from it in explanatory analyses.
>
> [As the authors claim that contextual embeddings …] The mixing weights in contextual embeddings are calculated as derivatives of head outputs to input embeddings. Because the non-identifiable attention weights will not influence any head outputs, they do not change the mixing weights. So the non-identifiable attention weights will not affect any mixing component of the contextual embeddings.
> We do not evaluate how the identifiability of tokens affects performance. We provide a tool and an analysis to increase our understanding of self-attention based models which in turn can help further research on how to improve these models.
>
> [Towards the claim that “non-linear activations in preserving token identity”...] Thanks for the suggestion. We have carried out said experiments for all combinations of model architecture (linear/MLP) and loss/measure (L2/Cosine) and added them to Appendix B5. We refer to the answers to reviewers #1 and #3 for an explanation of the newly introduced, cosine distance based experiments.
> TL;DR [start]: Both linear and non-linear models are able to identify most tokens (~92% in last layer on MRPC) when the nearest-neighbour lookup is done using cosine distance instead of L2. This suggests that identity is mostly encoded in the angle/orientation of the embeddings, while the magnitude/norm of embeddings might further encode “additional information” (such as aggregated context), as suggested by the lower identifiability rate when using L2 as a measure. [end]
> The experiments added to Appendix B5 show that g can be trained to identify neighbouring tokens to different degrees (L2 vs. cosine distance). For all experiments, the following trends hold: Identifiability rate quickly drops with distance in the sequence from the investigated token (input to g), and also decreases in later layers.
> Interestingly, identifying left vs. right neighbour tokens shows an asymmetric pattern, despite the bi-linear masked language modeling pre-training.

---

> > ### Comment · AnonReviewer2 · 2019-11-15
> > **Respond to authors**
> >
> > Thanks for your careful clarification, and thanks for updating the appendix and related work.
> >
> > I want to clarify that I understand the goal of your work.
> > I know your study focuses on interpreting the mode's behavior, providing tools and analysis to increase the understanding of the self-attention and also the transformer model.
> > I also agree with the findings you presented in the paper.
> >
> > Most of my concerns, seem to be beyond the scope of the paper, actually are not trying to "challenge" the correctness of your very interesting findings.
> > Overall, I would like to say:
> > Attention weight to some extent can not be used to identify the output, thus we should be careful when using it to interpret the model outputs.
> > However, such a kind of "uncertainty" seems to diminish when we look at contextual representation and the prediction of the model.
> > If a transformer-based model can (usually) achieve good performance, this means our training technique figures out a way to map to correct labels (and maybe stable contextual embedding in deep layers) though the attention weight is non-identifiable.

---

> > > ### Author Response · Authors · 2019-11-15
> > > **Response to reviewer**
> > >
> > > Thanks for the clarification and constructive comments.
> > > We think that you summarized the conclusions of our paper well.

---

### Official Review · AnonReviewer1 · 2019-10-23
**Official Blind Review #1**

**Rating:** 8

**Review:**

This work studies the identifiability of attention, i.e., to what extent does the attention weight uniquely determine the output. The paper (1) formally establishes, using a rank argument, that at least for the cases where the attention head dimensions are smaller than the sources, infinite different attention weights can yield the same attention output. Based on such an observation, (2) a principled tool to extract the `effective` attention is introduced, with which several previous observations are challenged. This tool can certainly inspire future research. The paper further (3) explores how much token and context information is mixed at different layers of a transformer model, using both a carefully designed probing task and a gradient-based method.

Overall I like the paper a lot. Its hypothesis and arguments are very clearly presented, which are then fleshed out by carefully designed experiments. I vote for an acceptance.

I don't have any major complaint. Below are some questions and comments.

- Since the paper only experimented with BERT, I think the authors mean "BERT" instead of the transformer models in general, whenever an empirical argument is drawn.

- Section 4.1, can the authors justify the use of MRPC for this analysis? Adding onto this, do the authors think the conclusion here holds for other input text domain? What about other transformer models, e.g., GPT, or a transformer trained using MT objective?

- Section 4.2, 2nd paragraph. I'm not sure why that g^{MLP} is better than g^{lin} can be interpreted as "non-linearities play an important role in maintaining token identifiability." Could it be the case that the non-linearities make the token information more opaque, such that it can only be extracted with a more expressive model (MLP)? A better way to support this claim, in my opinion, is to show a better token identifiability in a BERT model trained without non-linearities.

- In section 4.2, many conclusions are drawn that the last layer behaves differently than the rest. To eliminate the potential effect of depth, have the authors considered conducting a controlling experiment, where the probing task is to recover e_{i}^{l-1} (i.e., the representation from layer l-1, instead of the input token embedding) from e_{i}^{l}?


**Experience Assessment:**

I have published one or two papers in this area.

**Review Assessment: Checking Correctness Of Derivations And Theory:**

I carefully checked the derivations and theory.

**Review Assessment: Checking Correctness Of Experiments:**

I carefully checked the experiments.

**Review Assessment: Thoroughness In Paper Reading:**

I read the paper at least twice and used my best judgement in assessing the paper.

---

> ### Author Response · Authors · 2019-11-11
> **Response to Reviewer #1**
>
> Thanks for the careful examination of our work and the constructive feedback. We believe there is a lot more to investigate in this topic, beyond the scope of a single paper, and thus, we hope this will inspire more research. Regarding the questions and comments, next, we provide an answer to each of them:
>
> [Since the paper only experimented with BERT…] Thanks for pointing the need to specify that the empirical results correspond to BERT in particular. We have reviewed the manuscript to make sure that our phrasing is precise in this regard.
>
> [Section 4.1, can the authors justify…] Since the submission date we have extended our experiments to two more datasets from the GLUE benchmark: CoLA and MNLI-matched. These datasets are significantly larger than MRPC, and the examples have a different structure to those in MRPC. The results on these datasets present the same patterns as those on MRPC and are included in Appendix D in the new version of the manuscript. Therefore, we are confident that our conclusions generalize across data domains.
>
> [...What about other transformer models,...] We did not experiment with other transformer models and although our results are likely to generalize also in this dimension, an extensive evaluation and comparison of different models with the methods we propose in this work is an interesting avenue for future work.
>
> [... transformer trained using MT objective?] We like the idea of looking at generative language modeling, beyond text encoding and classification. In particular, MT, seems an obvious direction for future work since it provides a more solid framework for evaluating input-output correspondences.
>
> [Section 4.2, 2nd paragraph…] We thank you for raising this point, which is in line with the comments of Reviewer #3. We see that our experiments do not provide sufficient evidence to support this claim. We have analyzed this more thoroughly. Although training a BERT without non-linearities would probably be the best way to study the role of non-linearities, due to limited time we cannot run this experiment within the reviewing period and leave this for future work. However, we have conducted additional experiments to retrieve token identity, including using cosine distance instead of L2 distance, which lead to a different conclusion. Using cosine distance, we found that there is almost no difference between a linear perceptron and an MLP. The results further suggest that the identity of the embeddings is encoded in their angle/orientation and that it can be recovered using a simple linear transformation of the type g(x) = Wx. This can be seen from significantly higher identifiability rates when using cosine distance as compared to L2. While providing additional insight into the nature of token identity, these new experiments still support the main argument of our paper: Tokens are largely identifiable throughout BERT, while identifiability decreases with depth (albeit to a lesser extent than previously found). Our new version of the manuscript updates these points accordingly.
>
> [In section 4.2, many…] Thank you for suggesting this experiment. We have carried it out and our results show that the tokens can be mapped back from one layer to the previous one almost perfectly. This is not surprising since the task is strictly easier than mapping back to the input. Also, the last few layers (especially layer 12) seem to behave differently here as well, which provides evidence that the strange behaviour of layer 12 is not simply due to depth (i.e., due to some cumulative effect). These new results have been included in Appendix B4 in the new version of the manuscript.
>
> Overall, the hypothesis about the strange behavior of layer 12 is that training objectives that operate on a sentence-level (e.g., next sentence prediction, most downstream fine-tuning tasks) as opposed to token-levels (such as masked LM) cause the last layer(s) to behave differently to the earlier layers from the perspective of token identifiability. To investigate this further we also added experiments for two fine-tuned versions of BERT (MRPC, CoLA) to Appendix B5. These new experiments provide further evidence for the hypothesis above.

---

### Official Review · AnonReviewer3 · 2019-10-24
**Official Blind Review #3**

**Rating:** 6

**Review:**

CONTRIBUTIONS:
Topic: Analysis of self-attention in the standard Transformer, applied to models of language.
Notation for review (see also definitions in C1-C4): cell(t,L) is the cell in layer L at input-string-position t (“corresponding to input token t”); Z(t,L) is the output vector for cell(t,L). Also, in the review, points are sometimes labeled by “(label)”, which is intended to label the point that immediately *follows* it in the text.
C1. “Attention identifiability.” Uniquely identifying the attention-weight vector from the output of attention is not possible when D < T: For an individual head, the vector R returned by attention uniquely determines the attention-weight vector A only if the dimension D of the value vector equals or exceeds the length T of the input.
C2. “Effective attention vector” A1: When D < T, there is a method for uniquely decomposing A as A = A1 + A2 where A2 has no effect on R.
C3. “Token identifiability” method M. Identifying t from Z(t,L): A method M is proposed for this, and experimental results presented showing that, roughly, M succeeds at lower levels L but can fail at higher levels.
C4. “Hidden token attribution” method: A gradient-attribution-based method is proposed to quantify C(t,t’), the proportional contribution of input token t’ to Z(t,L). C(t,t) is the “[same-]token contribution”; the “contextual contributions” are the C(t,t’) for t’ =! t. Experimental results are presented showing that after L = 1, 6, 12 the median token values are about 30%, 15%, 10%.
MAIN ARGUMENT: Interpretation of the weight of cell(t,L)’s attention to cell(t’,L) as a measure of the influence of token t’ on the level-L representation of token t is (P0) frequent, but problematic, because (P1) the attention weight itself includes an irrelevant quantity (the A2 contribution, which should be omitted to get the effective attention [C2]) and because (P2) Z(t,L) is not properly viewed as representing token t, since (P2) it can be insufficient to even determine which token t it corresponds to [C3] as (P3) it can contain strong contributions from other tokens [C4].
RATING: Weak accept
REASONS FOR RATING (SUMMARY). The paper raises important challenges to a standard practice for interpreting the knowledge in Transformer models (especially BERT) by examining attention weights. It proposes methods for reaching more theoretically justified interpretations. It introduces important new general interpretive concepts in the process. The main argument is sufficiently well supported by theoretical and empirical results to justify making the ICLR community aware of it, although there is a missing step in the argument: justification that the proposed means of resolving the unidentifiability of attention yields an optimal result.
REVIEW
Weaknesses.
Minor complaint concerning exposition: The central result, C1 above, is actually rather obvious, unless I’m missing a subtlety (in which case it would be useful for that subtlety to be pointed out explicitly in the paper). (*) The dimension-D vector R is a linear combination of T vectors; if T > D, these T vectors cannot be linearly independent, so the weighting coefficients that yield R cannot be unique. The argument presented is rather overblown and may hide the obvious correctness of the result. However, I think over-deriving is clearly preferable to under-deriving, so this complaint truly is extremely minor. It might be worthwhile to add a sentence like (*) however to make sure that simple observation doesn’t get lost in the formalism. A further advantage of the more detailed derivation is that it sets up nicely the unique decomposition of A into the component A2 in the null space and the component A1 orthogonal to the null space (of the matrix containing the T vectors being linearly combined).
A more substantial question here: What argument or evidence is there that this particular decomposition A = A1 + A2  is the one that yields a definition of “effective attention” that is best for interpretative purposes? Is there a metric of success that would validate this? Given that A is underdetermined, there is a whole subspace of attention-weight vectors that yield the same result R of attention. The proposed choice of A1 within that space is a natural one, but what makes it the *right* one? The incontrovertible point is that the A vector a model happens to use has a degree of arbitrariness in it that makes using it for interpretation problematic. But what is the right criterion for picking an alternative “effective” version of A, A1? Being orthogonal to the null space is one well-defined choice, but is it clearly the right choice? (I believe this choice is also the one with minimal L2 norm. Is that important?) For example, Sec. 3.3 shows that raw attention, but not effective attention, gives evidence that attention shifts from [CLS] to [SEP] to periods and commas as L increases. This shows they are different, but what is the argument that the raw-attention conclusion is less ‘correct’ than the effective-attention conclusion here?
More generally, it would strengthen the paper even further to document more concrete cases where effective attention yields different interpretive conclusions than raw attention, especially to the extent that it can be argued that the former interpretations are somehow better than that latter.
Strengths.
The paper gives convincing evidence of the MAIN ARGUMENT’s propositions (P0) – (P3).
(P0) [Interpretation of the attention weight as a measure of the influence of one token on another] is frequent] Sec 4, 2nd paragraph, cites 16 papers to illustrate the point. While I have not examined all of them, even allowing for some possible disagreements of interpretation, I am confident there is ample relevant evidence among these papers and probably others as well.
(P1) [the attention weight itself includes an irrelevant quantity [C2]] The argument provided for this in Sec 3 seems correct to me, but as stated above as a ‘minor complaint concerning exposition’, it may not make as clear as possible the obviousness of the conclusion.
(P2) [Z(t,L) can be insufficient to determine which token t it corresponds to [C3]] This point is operationalized in Sec 4.2 by picking a function-approximation architecture G, and attempting to use it to perform the map Z(t,L) -> t, for a given L and for all input strings and tokens t. A linear and a non-linear MLP G are used. An experiment applying the pre-trained BERT-base Transformer to the Microsoft Research Paraphrase Corpus gives success rates that fall to 73% and 82% by final layer 12. That 18% of tokens cannot be recovered from their corresponding layer-12 encoding is the basis of the claim that equating attention from cell(t,L) to cell(t’,L) with attention from token t to token t’ is not justified in general.
That the non-linear approximator does 9% better than the linear one is the basis of the claim that ‘the non-linearities in BERT play an important role in maintaining token identifiability’ (Sec. 4.2). This claim should be spelled out explicitly. It reads to me to say that if the non-linearities in BERT were removed, token identifiability would be reduced, but the experiment does not show that. Rather, it shows that, in the presence of the non-linearities in BERT, a linear approximation to the token-identifying map G is significantly less accurate than the non-linear approximation: hardly a surprising conclusion. This bit of the argument needs to be clarified.
(P3) [Z(t,L) can contain strong contributions from tokens t’ =! t [C4]] As defined in Sec. 5, the contribution of token t’ to Z(t,L), for a given L, is proportional to the norm of the gradient of Z(t,L) with respect to the input from token t’; these are normalized to sum to 1. This is a transparent definition that resembles the previous gradient attribution proposal, but applied to internal rather than output representations. Using the same experimental set-up, it is shown that the average contribution of token t to Z(t,L) declines from 30% to 10% as L goes from 1 to 12. The proportion of values t such that the largest contribution to Z(t,L) is from a token t’ =! T rises from 0 for L = 1 through 3 to over 30% for L = 10 through 12. This is further evidence for the claim that equating attention between Z(t,L) and Z(t’,L) with attention between tokens t and t’ is problematic.

**Experience Assessment:**

I have published one or two papers in this area.

**Review Assessment: Checking Correctness Of Derivations And Theory:**

I carefully checked the derivations and theory.

**Review Assessment: Checking Correctness Of Experiments:**

I carefully checked the experiments.

**Review Assessment: Thoroughness In Paper Reading:**

I read the paper thoroughly.

---

> ### Author Response · Authors · 2019-11-11
> **Response to Reviewer #3**
>
> Many thanks for the in depth evaluation and review of our work, we appreciate the feedback. We addressed the comments and questions, improving the paper in this way. Please find below our answers to the weaknesses you address:
>
> Regarding C1, we have gladly included a sentence along the line of (*) in the derivation. Please, refer to the updated manuscript after Eq. 2, [Intuitively, the head output…]
>
> [A more substantial question here:...] The study of attention identifiability focuses on interpreting a model’s behavior. Here the main argument for the decomposition of attention raw(A) = effective(A)+null(A) is to distinguish the part of attention influencing the model output, effective(A), from the non-influencing part, null(A). The argument is presented in terms of the basic linear algebra fact that null(A) cannot influence downstream model computations, i.e.,: it does not affect the solution b to Ax=b, since Ax=b = effective(A)x + null(A)x = effective(A)x = b, because null(A)x = 0. We agree that this appears obvious, in retrospect. However, it had not been noticed before and it might inform further research in this direction.
>
> We clarify that we don’t mean “effective” as better, correct or optimal. Further work is needed to properly compare, quantitatively and qualitatively, this proposal. However it seems safe to conclude that interpretations based on null(A) might overlook important findings or overinterpret spurious factors. Thus, we mean “effective” just in the sense that it actually “affects”, or “has an effect” on,  the computation of the output. The name “effective” is meant to emphasize this.
>
> [...minimal L2 norm. Is that important?] In addition, as you pointed out, effective attention, which is defined as the orthogonal component to the null-space, is indeed an equivalent attention with minimal L2 norm. Because any equivalent attention = effective(A) + anyother_null(A). Its norm^2 =norm(effective(A))^2+norm(anyother_null(A))^2>=norm(effective(A))^2. The equal sign holds if anyother_null(A)=0. So the minimal L2 norm equivalent attention is the effective attention effective(A). As these two definitions are mathematically equivalent, the orthogonal definition is adopted for simplicity.
>
> For the example mentioned, Clark et al. [1] show that the middle layers attend to the [SEP] token (as reproduced in our Figure 1a). They follow up with a gradient based investigation and conclude that this attention to the [SEP] token is generally a 'no-op'. Effective attention can provide more consistent suggestions that this pattern (raw attention peak on [SEP]) is irrelevant to the computation of the output for middle layers. Furthermore, the effective attention can tell if some attention on [SEP] may have some effect on head outputs. The same argument works for the sharp peak of raw attention on tokens [.] or [,] between layers 10 and 12 in [1] (also reproduced in Figure 1a).
>
> With respect to the suggestion of documenting more concrete cases where effective attention yields different interpretive conclusions, we have added a section in the appendix (A.3) where we analyze one such case in more depth. We find that raw and effective attention overlap to a significant degree, e.g., near the diagonal components of the attention matrix, but display also complementary views. In particular, we found that removing the null component can highlight subtle long distance dependencies that are obfuscated by the null attention.
>
> [...claim that ‘the non-linearities in BERT play an important role in...] We thank you for raising this point, which is in line with the comments of Reviewer #1. We see that our experiments do not provide sufficient evidence to support this claim. We have analyzed this more thoroughly. A thorough investigation in this matter would require retraining BERT without the non-linearities (as suggested by R#1), which we leave for future work due to time constraints. However, we have conducted additional experiments to retrieve token identity, including using cosine distance instead of L2 distance which lead to a different conclusion. Using cosine distance, we found that there is almost no difference between a linear perceptron and an MLP. The results further suggest that the identity of the embeddings is encoded in their angle/orientation and that it can be recovered using a simple linear transformation of the type g(x) = Wx. This can be seen from significantly higher identifiability rates when using cosine distance as compared to L2. While providing additional insight into the nature of token identity, these new experiments still support the main argument of our paper: Tokens are largely identifiable throughout BERT, while identifiability decreases with depth (albeit to a lesser extent than previously found). Our new version of the manuscript updates these points accordingly.
>
> References:
> [1] K. Clark, U. Khandelwal, O. Levy, and C. D. Manning. What does BERT look at?an analysis of bert’s attention, 2019.

---

> > ### Comment · AnonReviewer3 · 2019-11-15
> > **AnonReviewer3**
> >
> > I have read the reviews and responses (thank you, authors!) and on the basis of those comments and the numerous clarifications and substantial additions to the paper I am raising my score to 8: Accept.

---

> > > ### Author Response · Authors · 2019-11-15
> > > **Response to reviewer**
> > >
> > > Thanks again for helping us improve the paper.

---

### Decision · Program_Chairs · 2019-12-19

**Decision:**

Accept (Poster)

**Comment:**

This paper investigates the identifiability of attention distributions in the context of Transformer architectures. The main result is that, if the sentence length is long enough, difference choices of attention weights may result in the same contextual embeddings (i.e. the attention weights are not identifiable). A notion of "effective attention" is proposed that projects out the null space from attention weights.

In the discussion period, there were some doubts about the technical correctness of the identifiability result that were clarified by the authors. The attention matrix A results from a softmax transformation, therefore each of its rows is constrained to be in the probability simplex -- i.e. we have A >= 0 (elementwise) and A1 = 1. In the present version of the paper, when analyzing the null space of T (Eqs. 4 and 5) this constraint on A is not taken into account. In particular, in Eq. 5 the existence of a \tilde{A} in the null space of T is not clear at all, since for (A + \tilde{A})T = AT to hold we would need to require, besides A >= 0 and A1 = 1, that A + \tilde{A} >= 0 and A1 + \tilde{A}1 = 1, i.e.

\tilde{A} <= A (elementwise)
\tilde{A}1 = 0

The present version of the paper does not make it clear that the intersection of the null space of T with these two constraints is non-empty in general -- which would be necessary for attention not to be identifiable, one of the main points of the paper.

The authors acknowledged this concern and provided a proof. I suggest the following simplified version of their proof:

We're looking for a vector \tilde{A} satisfying

(1) \tilde{A}’*T = 0 (to be in the null space of T)

and

(2) \tilde{A}’*1 = 0
(3) \tilde{A} >= -A

(to make sure A + \tilde{A} are in the probability simplex).

Conditions (1) and (2) are equivalent to require \tilde{A} to be in the null space of [T; 1]. It is fine to assume this null space exists for a general T (it will be a linear subspace of dimension ds - dv - 1).

To take into account condition (3) here’s a simpler proof: since A is a probability vector coming from a softmax transformation (hence it is strictly > 0 elementwise), there is some epsilon > 0 such that any point in the ball centered on 0 with radius epsilon is >= -A.

Since the null space of [T; 1] contains 0, any point \tilde{A} in the intersection of this null space with the epsilon-ball above satisfies (1), (2), and (3). This should work for any ds - dv > 1  and as long as A is not a one-hot distribution (otherwise it collapses to a single point \tilde{A} = 0).

I am less convinced about the justification to use an “effective attention” which is not in the probability simplex, though (not even in the null space of [T; 1] but only null(T)). That part deserves more clarification in the paper.

I recommend acceptance of this paper provided these clarifications are provided and the proof is included in the final version.

---

> ### Author Response · Authors · 2020-02-10
> **Updated Camera Ready Version**
>
> Thank you for the critical review and insights that helped us to further improve the paper. We included a section into the camera ready version discussing the non-identifiability in the probability simplex. We also revised the other parts of the paper to reflect our insights as accurately as possible and look forward to fruitful discussions at the conference.